# EMONET-FACE: An Expert-Annotated Benchmark for Synthetic Emotion Recognition

**Christoph Schuhmann**[1,2*] **Robert Kaczmarczyk**[1,3*] **Gollam Rabby**[2]

**Felix Friedrich**[4,5] **Maurice Kraus**[4] **Krishna Kalyan**[1] **Kourosh Nadi**[1]

**Huu Nguyen**[1,6] **Kristian Kersting**[4,5,7] **Sören Auer**[2,8]

[1]LAION e.V., [2]L3S Research Center and Leibniz University of Hannover,
[3]Technical University of Munich, [4]TU Darmstadt, [5]Hessian.AI, [6]Ontocord, [7]DFKI,
[8]TIB, Leibniz Information Centre for Science and Technology

## Abstract

Effective human-AI interaction relies on AI's ability to accurately perceive and interpret human emotions. Current benchmarks for vision and vision-language models are severely limited, offering a narrow emotional spectrum that overlooks nuanced states (e.g., bitterness, intoxication) and fails to distinguish subtle differences between related feelings (e.g., shame vs. embarrassment). Existing datasets also often use uncontrolled imagery with occluded faces and lack demographic diversity, risking significant bias. To address these critical gaps, we introduce EMONET-FACE, a comprehensive benchmark suite, featuring: (1) A novel 40-category emotion taxonomy, meticulously derived from foundational research to capture finer details of human emotional experiences. (2) Three large-scale, AI-generated datasets (EMONET-FACE HQ, EMONET-FACE BINARY, and EMONET-FACE BIG) with explicit, full-face expressions and controlled demographic balance across ethnicity, age, and gender. (3) Rigorous, multi-expert annotations for training and evaluation. (4) We build EMPATHICINSIGHT-FACE, models that achieve human expert performance on our held-out benchmark and demonstrate strong real-to-sim generalization. The publicly released EMONET-FACE suite—taxonomy, datasets, and model—provides a robust foundation for developing and evaluating AI systems with a deeper understanding of human emotions.

## 1 Introduction

The advent of highly capable Large Language Models (LLMs) such as ChatGPT [34] has transformed human-computer interaction, a transformation further accelerated by advances in voice synthesis and multimodal reasoning [35]. Modern voice interfaces now enable emotionally expressive AI conversations [45, 25], leading to heightened expectations for AI systems with genuine emotional intelligence. As AI platforms like Replika [29], Character.ai [9], and Kajiwoto [22] are increasingly used for companionship and support [10], and as LLMs are consulted for mental health advice [48, 23, 16, 47], it becomes ever more critical for these systems to accurately interpret nonverbal emotional cues, especially facial expressions. Misreading users' emotions can erode trust and have negative psychological repercussions [8].

---

*Contributed equally and jointly supervised, correspondance to `contact@laion.ai`

39th Conference on Neural Information Processing Systems (NeurIPS 2025) Track on Datasets and Benchmarks.

Table 1: Comparison of facial emotion recognition datasets, including synthetic generation and expert annotation. Open license means CC-BY 4.0 or similar.

| Dataset Name | Open License | Size | Emotions | Synthetic | Expert Labels | Continuous | Diversity |
|---|---|---|---|---|---|---|---|
| FERG-DB [2] | ✗ | 55,767 | 7 | ✗ | ✗ | ✗ | ✗ |
| FERG-3D-DB [1] | ✗ | 39,574 | 7 | ✗ | ✗ | ✗ | ✗ |
| SZU-EmoDage [19] | ✓ | 840 | 8 | ✓ | (✓) | ✗ | ✗ |
| EmotionNet [17] | ✗ | N/A | 6 | ✗ | ✗ | ✗ | N/A |
| EMOTIC [24] | ✗ | 18,313 | 26 | ✗ | ✗ | (✓) | (✓) |
| FindingEmo [51] | ✓ | 25,869 | 8 | ✗ | ✗ | ✓ | (✓) |
| AffectNet [33] | ✗ | 450,000 | 8 | ✗ | ✗ | ✗ | (✓) |
| **EMONET-FACE BIG** | ✓ | 203,201 | 40 | ✓ | ✗ | ✗ | ✓ |
| **EMONET-FACE BINARY** | ✓ | 19,999 | 40 | ✓ | ✓ | ✗ | ✓ |
| **EMONET-FACE HQ** | ✓ | 2,500 | 40 | ✓ | ✓ | ✓ | ✓ |

At the same time, as AI systems become more embodied—through avatars and realistically synthesized faces—another crucial challenge emerges: enabling humans to correctly perceive and interpret the emotional states of these digital agents. For emotionally rich and affective interaction, the expressions generated by machine avatars must be legible and interpretable to human users, ensuring users understand not only the intentionality but also the emotional stance and nuance behind an AI's responses. This reciprocity is critical for building trust, rapport, and effective communication in human-AI partnerships. Truly emotionally aware AI thus requires a bidirectional capability: machines must recognize not only basic emotions [13] but also complex and subtle affective states [36, 12, 39], while synthetic emotions generated by AI avatars must be expressively rich and clearly understood by human counterparts. However, current benchmarks are limited in size, their spectrum of emotions, the diversity of groups included, and the quality of their annotations, constraining progress toward AI systems capable of authentic, context-sensitive empathy and expressiveness [11].

To address these gaps, we introduce EMONET-FACE, an expert-annotated data suite for fine-grained facial emotion recognition (FER) grounded in a novel, expansive 40-category emotion taxonomy. This taxonomy, developed by mining the "Handbook of Emotions"[27] and refined with psychological expertise, captures a broader and more detailed array of human emotional states than previous efforts. Using state-of-the-art text-to-image models [6, 32], we generated a series of structured, demographically balanced, high-quality synthetic datasets, covering a wide range of age, ethnicity, and gender. This includes a pretraining set of over 203,000 images (EMONET-FACE BIG), a fine-tuning dataset (EMONET-FACE BINARY) with almost 20k images and more than 62,000 human expert binary emotion annotations, and an evaluation benchmark (EMONET-FACE HQ) of 2,500 images meticulously rated by psychology experts using continuous scales across all 40 emotion categories. Alongside these datasets, we present EMPATHICINSIGHT-FACE, two models trained on our suite that achieve human-expert-level performance on the EMONET-FACE HQ benchmark and outperform concurrent models.

Our main contributions are as follows: **(i)** We introduce a comprehensive 40-category emotion taxonomy, refined through expert consultation, to capture fine-grained human emotional states; **(ii)** We construct diverse synthetic expert-annotated image datasets—spanning pretraining, fine-tuning, and benchmarking test sets—using state-of-the-art text-to-image models; **(iii)** We develop EMPATHICINSIGHT-FACE models that match human performance on fine-grained emotion recognition; **(iv)** We openly release our datasets and models to foster research on emotions in AI.[2].

## 2 Related Work

Understanding human emotion remains a complex task, grounded in diverse psychological theories and demanding robust data for computational modeling. Emotions are challenging as they are not universal—they are socially constructed, shaped by cultural norms and context (further elaborated in

---

[2]We openly release our datasets, models, training scripts, and open-source annotation tool at our HuggingFace and GitHub repositories. Generation prompts are included in the dataset metadata.

[3, 4] and App. A.1). This section outlines key theoretical frameworks, examines existing emotion taxonomies and datasets, and identifies the specific gaps that EMONET-FACE addresses.

**Existing Emotion Taxonomies.**   Theories of emotion span a spectrum from universal, biologically rooted models to constructivist perspectives, with the latter emphasizing that emotions are shaped by context and culture rather than being innate, static programs [3, 4]. This conceptual diversity has direct consequences for how emotions are categorized and labeled in practice. Traditional taxonomies provide important foundations but often lack granularity. Ekman's model expanded to include emotions like amusement and shame [14], but remains limited to discrete, universal states. Parrott's hierarchical structure [38] and Plutchik's Wheel [40] offer more coverage but overlook distinctions critical for real-world applications. For example, Plutchik's dyadic model emphasizes emotional opposites and blends but may oversimplify ambiguous affective states. Other approaches include Panksepp's neurobiologically rooted systems (e.g., SEEKING, FEAR, LUST) [37], and Izard's emphasis on distinct expressive and neural signatures for emotions like joy, guilt, and interest [20, 21]. Recent efforts have enriched positive emotion taxonomies—e.g., Shiota et al.'s physiology-based framework [46] and Weidman and Tracy's focus on subjective feeling states [49]. Despite these contributions, existing emotion taxonomies often omit physical states and stances like fatigue, pain, numbness, or sourness, and do not clearly distinguish closely related states such as gratitude, affection, and lust. EMONET-FACE addresses these gaps with a broader, literature-informed taxonomy [27] expanding basic emotions with relevant physical states and stances to a 40-category taxonomy, tailored for high-resolution FER.

**Emotion Recognition Datasets.**   Existing emotion recognition datasets vary significantly in terms of realism, diversity, and labeling accuracy. Early posed datasets (e.g., CK+ [28], JAFFE [30]) rely on actor-performed facial expressions, offering clear labels but limited real-world validity. In-the-wild datasets (e.g., AffectNet [33], EmotioNet [5], EMOTIC [24], FindingEmo [51]) provide more realistic expressions with contextual richness. However, they suffer from confounders like context-dependent perception effects (e.g., Kuleshov effect [7], background assimilation [50]), as well as occlusions and inconsistent image quality. In contrast, EMONET-FACE overcomes these limitations by offering high-resolution, synthetic facial images rendered under controlled conditions. To this end, our dataset offers deliberate and precise control over demographic diversity and background context for every emotion, in contrast to previous datasets, which might achieve some diversity incidentally by web scraping. Additionally, it is unique in offering expert annotations across 40 emotion categories. A detailed comparison with previous datasets is presented in Table 1.

## 3   Building EMONET-FACE: Fine-grained Expert Emotion Datasets

Having identified several critical gaps in existing emotion datasets—including limited taxonomic granularity, insufficient demographic and contextual diversity, image quality, and a lack of expert annotations—we turn next to how we constructed EMONET-FACE to address these challenges.

**Introducing a Fine-grained Taxonomy.**   To enable more fine-grained affective understanding in AI, we developed a 40-category emotion taxonomy grounded in contemporary psychology and informed by the Theory of Constructed Emotion (TCE) [4]. Unlike prior work limited to basic emotions such as joy, anger, fear, sadness, and disgust, our taxonomy intentionally extends to a broad spectrum of fine-grained social, cognitive, and bodily states—including elation, contentment, gratitude, hope, pride, interest, awe, astonishment, relief, longing, teasing, embarrassment, shame, disappointment, contempt, sexual lust, doubt, confusion, envy, jealousy, bitterness, pain, fatigue, numbness, helplessness, and less common categories such as sourness and intoxication. Each category encompasses a cluster of semantically associated descriptive words, systematically extracted from the *Handbook of Emotions* [27] and refined via expert consultation (see App. Table 5 for full taxonomy).

To construct this taxonomy, we digitized the 946-page *Handbook of Emotions* via OCR, segmented the text into 500-word blocks, and used GPT-4 to extract candidate emotion nouns. After aggregation and deduplication, we identified 170 unique terms, which were then clustered through iterative rounds of independent listing, critical review, and expert-guided refinement with psychologists and researchers. Aligned with TCE, we do not claim biological universality; instead, our taxonomy is designed for context-aware, socially informed emotion interpretation in AI. Given the inherent ambiguity in facial emotion perception—e.g., high-arousal expression might plausibly reflect amusement, elation, or

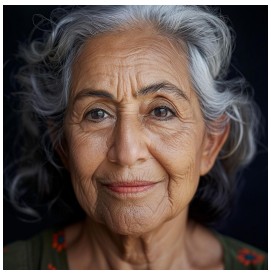 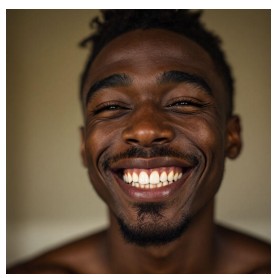 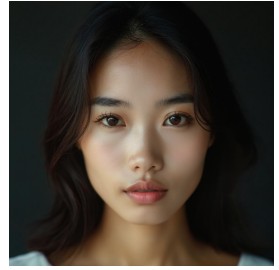

(a) **Prompt:** An authentic, realistic closeup image of a Hispanic or Latino woman of 70 years who seems to experience triumph, superiority. Strong facial expression of triumph, superiority, [...] *(Midjourney v6)*

(b) **Prompt:** A focused close-up photo of a Black or African American man of 30 years who seems to genuinely experience lighthearted fun, amusement, mirth, joviality, laughter, playfulness, silliness, [...] *(Flux Pro)*

(c) **Prompt:** A closeup sharp, focused photo of a Southeast Asian woman of 40 years who seems to genuinely experience mild yearning, longing, pining, wistfulness, nostalgia, Craving, desire, Envy, [...] *(Flux Dev)*

Figure 1: Samples from our EMONET-FACE datasets generated with different sota T2I models.

excitement—we designed the taxonomy for plausible multi-label annotations over rigid single-label assignments, supporting richer, context-aware emotion representations.

**Image Generation and Selection Methodology.** To construct the EMONET-FACE datasets, we generated synthetic facial images representing all 40 emotion categories using a controlled prompt-engineering pipeline. Each image was manually screened for quality, ensuring visual consistency and rejecting those with artifacts or ambiguous expressions. Critically, prompt templates were explicitly designed to ensure demographic diversity by balancing gender identity (45% man, 45% woman, 10% non-binary), capturing a wide age range (20–80 years, in 10-year increments), and representing 14 different ethnic backgrounds (e.g., "Middle Eastern," "South Asian"). Exemplary generated images are depicted in Figure 1. In total, we generated more than 200k images. While labor-intensive, this manual expert curation was essential to create a gold-standard benchmark, as automated filtering cannot reliably detect the subtle artifacts or ambiguous expressions that would compromise the integrity of a dataset for fine-grained emotion recognition.

**Acquiring Expert Annotations.** To ensure precise high-quality emotion labels in EMONET-FACE, we engaged psychology experts recruited via Upwork, selected for their verified academic degrees. Our team of annotators (13 in total), aged 18–44 and spanning diverse ethnic and linguistic backgrounds—e.g., Hispanic, European, South Asian, and Middle Eastern—with fluency in up to four languages, brought a diverse perspective. Annotation was conducted on our open-source platform (see App. C). Annotators were instructed to follow our detailed guidelines, rather than relying solely on personal intuition, following [42, 43], promoting coherence. Details on annotator demographics, recruitment, compensation, and adherence to ethical guidelines are provided in App. A.2 and checklist.

For EMONET-FACE HQ, we used a continuous emotion rating protocol: each expert independently assessed batches of 250 images drawn from the 2,500-image collection, scoring all 40 emotion categories for every image on a 0–7 intensity scale. This multi-rater approach, with four ratings per image, supported fine-grained, multi-label assignments for the ambiguous and overlapping expressions commonly found in facial emotion data. To construct EMONET-FACE BINARY, we implemented a rigorous multi-stage binary annotation protocol. In the affirmative sequence, images initially identified as positive for a target emotion by one annotator were reviewed by a second, and if confirmed again, by a third, resulting in emotion labels with triple positive consensus. Images with any annotator dissent were excluded to ensure high-confidence positives. Additionally, a contrastive batch was included, in which annotators reviewed images with the target emotion deliberately absent, ensuring the inclusion of high-quality true negatives (see Table 2). This combination provides robust verification of both the presence and absence of emotion. For EMONET-FACE BIG, we synthetically annotated over 200k images in a two-stage process with Gemini-2.5-Flash. Initially, the model identified and scored the five most salient emotional dimensions per image; subsequently, to boost detection of underrepresented emotions, we used a "hinting" strategy that prompted the model to arbitrarily consider specific emotions. This automated, iterative process continued until we achieved broad coverage across all 40 categories in our taxonomy.

Table 2: Annotation protocol and agreement rates for EMONET-FACE BINARY. "Affirmative" batches required triple positive agreement across annotators, while the "contrastive" batch ensured true negatives. Agreement indicates the proportion of images for which annotators consistently agreed on presence (affirmative) or absence (contrastive) of the target emotion, which is high overall.

| Batch | # Images | Agreement | # Annotators | Step |
|---|---|---|---|---|
| Batch 1 (affirmative) | 19,999 (100.00%) | 82.83% | 3 | Starting point |
| Batch 2 (affirmative) | 14,670 ( 73.35%) | 75.11% | 4 | Batch 1 positives |
| Batch 3 (affirmative) | 11,018 ( 55.09%) | 82.37% | 4 | Batch 2 positives |
| Batch 4 (contrastive) | 19,999 (100.00%) | 94.29% | 4 | Batch 1 contrastive |

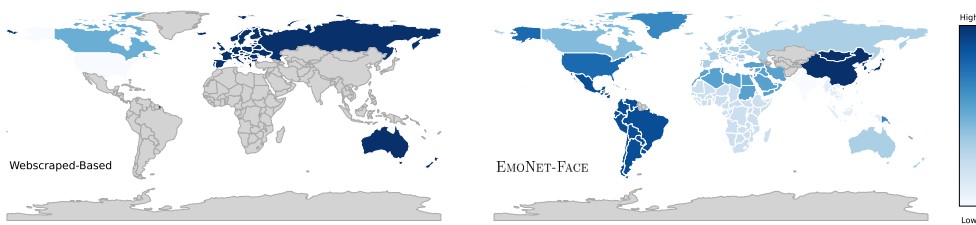

Figure 2: Approximate world map of demographic coverage and diversity in web-scraped datasets (left) compared to EMONET-FACE (right), which is much more diverse.

**Inter-Annotator Agreement.**   To quantify annotator consistency for EMONET-FACE, we evaluated agreement using both pairwise and group metrics for each emotion. For the continuous 0–7 scale in EMONET-FACE HQ, we used weighted kappa ($\kappa_w$, with quadratic weights) to capture how closely pairs of annotators rated images, considering only cases where both provided ratings. We also computed Krippendorff's $\alpha$ across all annotators for both EMONET-FACE HQ and EMONET-FACE BINARY. We also compute the conditional agreement for EMONET-FACE BINARY, following the sequential labeling, as described above.

The average pairwise $\kappa_w$ between human annotators for EMONET-FACE HQ was 0.20 (see Figure 3, A, top part), indicating moderate agreement. Furthermore, for EMONET-FACE HQ, we depict Krippendorff's $\alpha$ per emotion in Figure 9. Across all 40 emotions, the mean $\alpha$ is 0.19 (95% CI [0.16, 0.22]), indicating moderate overall agreement. Per-emotion $\alpha$ varies widely: highest for Elation ($\alpha = 0.58$), Amusement (0.56) and Anger (0.46); lowest (even negative) for Interest ($-0.08$), Concentration ($-0.02$) and Contemplation ($-0.02$). This spread suggests that stimulus ambiguity, not annotator error, drives most disagreement.

For EMONET-FACE BINARY, we get overall $\alpha = 0.09$ (95% CI [0.09, 0.09]), as depicted in Figure 8, with the top five by binary $\alpha$ are Fatigue/Exhaustion ($\alpha = 0.49$), Pain (0.45), Malevolence (0.44), Teasing (0.43) and Intoxication (0.40). For conditional binary (presence/absence) agreement, consensus rates are reported in Table 2. High agreement was achieved across batches, with especially strong consensus in the contrastive (true-negative) setting, reflecting the comparative ease of judging when an emotion is absent, compared to present.

Taken together, these results demonstrate that while human agreement is strong for many emotions, some categories naturally elicit a wider range of interpretations, underscoring the nuanced nature of facial emotion expressions. Rather than indicating weak annotation quality, this pattern highlights the sensitivity of EMONET-FACE to the inherent complexity of affective perception. Our annotations reflect both the challenges and scientific opportunities of capturing emotional diversity.

**Data Integrity, Safety, and Fairness.**   All generated images were manually reviewed for obvious stereotypes, artifacts, or harmful content. This way, one inappropriate image was removed from EMONET-FACE BINARY. The EMONET-FACE dataset is released for research under ethical terms of use, supporting safe and fair study of affective AI. Moreover, we focused on a broad and diverse representation across identities. The development of fair and unbiased FER systems critically depends on diverse training and evaluation datasets. While some existing datasets (see Table 1) include varied identities, equitable coverage across groups and emotional categories remains limited to date. In

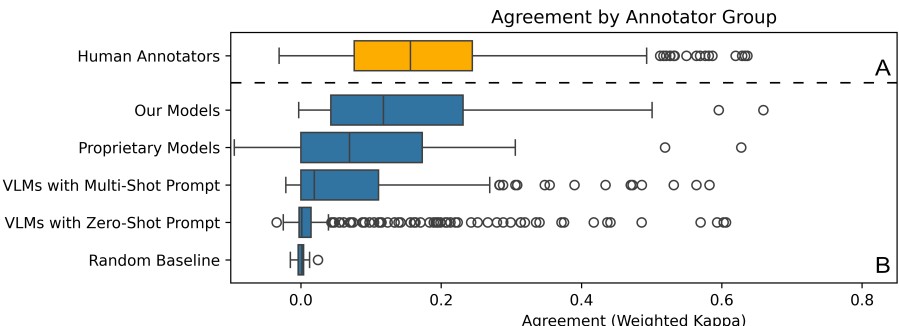

Figure 3: Weighted Kappa ($\kappa_w$) agreement scores by annotator group. **A** (top): Pairwise agreement between human annotators. **B** (below): Pairwise agreement between each human annotation and other sources, including 'Our Models' (EMPATHICINSIGHT-FACE), 'Proprietary Models' (HumeFace), 'VLMs (Multi-Shot and Zero-Shot Prompts)', and a 'Random Baseline'. Each box represents the interquartile range (IQR) of $\kappa_w$ scores, with the median as center line.

creating our EMONET-FACE datasets, we explicitly addressed this by focusing on diversity across ethnicity, gender, and age groups. Figure 2 illustrates the approximate demographic diversity between average web-scraped datasets, predominantly White/Caucasian [31], and our datasets, which are much less Western-centric and more diversified across identities. For additional visualizations, refer to the supplementary files.

### 3.1 The EMONET-FACE Suite

After having established and evaluated its foundation, we release EMONET-FACE as three subsets:

(1) **EMONET-FACE HQ**: a *benchmark test dataset* with 2,500 images with multiple continuous expert annotators per image, resulting in 10,000 human expert annotations;

(2) **EMONET-FACE BINARY**: a *fine-tuning dataset* with 19,999 images with multiple binary expert annotators per image from a multi-stage process, resulting in 65,686 human expert annotations;

(3) **EMONET-FACE BIG**: a *pre-training dataset* with 203,201 images and generated labels.

## 4 Benchmarking Facial Emotion Recognition on EMONET-FACE

We now use EMONET-FACE to benchmark FER models, including our baselines, proprietary systems, and VLMs, against human expert annotations.

### 4.1 Experimental Setup

**Developing Baseline Models: EMPATHICINSIGHT-FACE.**   In addition to establishing new FER datasets, we leverage them to train baseline models, which we term EMPATHICINSIGHT-FACE. Our architectural design is deliberately simple—a SIGLIP2 vision backbone followed by lightweight MLP regression heads—to demonstrate that strong performance can be achieved with a straightforward architecture when paired with high-quality data and robust embeddings. We chose continuous regression (predicting scores on a 0–7 scale) over classification for two key reasons: (1) *Capturing Intensity:* Regression naturally models the gradations of emotional expression, distinguishing mild amusement from intense elation, a nuance lost in discrete classification. (2) *Multi-Label Representation:* Faces often display blends of emotions. Our regression approach allows for predicting multiple emotions simultaneously (e.g., high sadness and moderate disappointment), providing a richer and more realistic affective profile. We pre-train on EMONET-FACE BIG and fine-tune on EMONET-FACE BINARY. Two model sizes are trained—EMPATHICINSIGHT-FACE LARGE (40 x 1.8M-parameter heads) and EMPATHICINSIGHT-FACE SMALL (40 x 151k-parameter heads). Full details in App. B.2.

**State-of-the-Art Models.**   To benchmark SOTA FER, we evaluated several open- and closed-source VLMs—including Gemini (various versions), Claude 3.7 Sonnet, GPT-4o, Nova Pro, Pixtral Large,

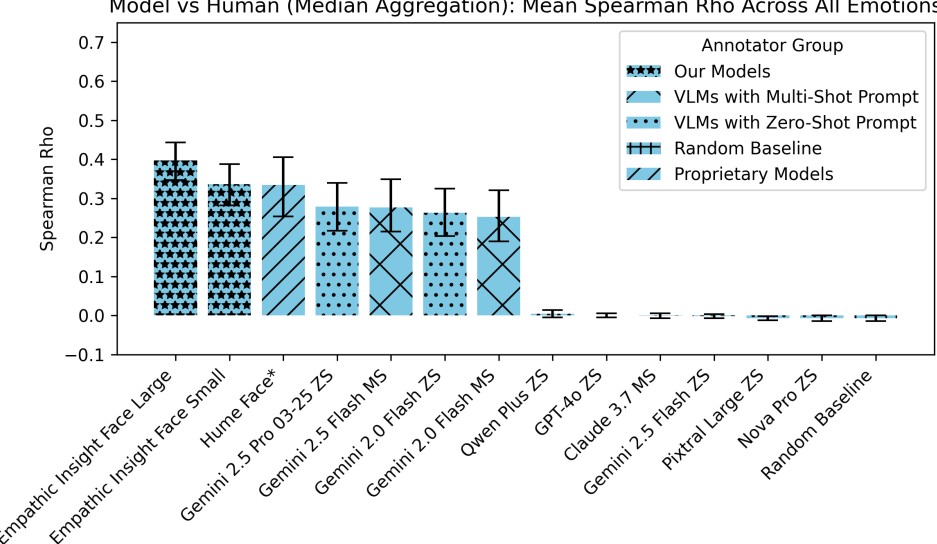

Figure 4: Mean Spearman's $\rho$ correlation between various model annotators and human annotations. Human ratings were median-aggregated per emotion before correlation with model ratings. The bar heights represent the mean of these per-emotion Spearman's $\rho$ values, calculated across all emotions for each model. Error bars indicate bootstrap 95% confidence intervals (N=1000 bootstraps) for these means. Model annotator groups, including our trained models (EMPATHICINSIGHT-FACE), VLMs with multi-shot or zero-shot prompting, proprietary models (HumeFace), and a random baseline, are distinguished by patterns as detailed in the legend

and Qwen Plus—on zero-shot and multi-shot prompt formats, and the proprietary HumeFace API. Model outputs were considered successful if they returned correctly formatted and parseable emotion labels; failures included refusals, empty responses, or unparseable outputs as reported in Table 3. The high failure rate of many SOTA VLMs is a significant finding of our benchmark, highlighting their brittleness for this task. For our performance analysis, any such failure was treated as a zero-agreement score, reflecting a complete inability to perform the task.

## 4.2 Do They See What We See?

We now examine how closely models—both specialized and general-purpose—recognize the same emotions as humans in synthetic faces, and how strongly their judgments align. To provide a clear and concise comparison across annotator groups, Figure 3 aggregates pairwise agreement scores (Weighted Kappa, $\kappa_w$) by model type and prompt format. This aggregation highlights several important trends: our models (EMPATHICINSIGHT-FACE) achieve consistently high agreement with human annotators; proprietary models (HumeFace) show only slight agreement; and, as expected, random baselines display negligible agreement. Most notably, VLM performance is highly variable and generally inconsistent—regardless of prompt format, VLMs on average do not offer a dependable improvement over random guessing. As illustrated in Table 3, this inconsistency is compounded by high rates of unparseable outputs or outright refusals, further hindering their reliability.

Strikingly, further statistical analyses (Pairwise Mann–Whitney U, bootstrapped 95% confidence intervals) show that EMPATHICINSIGHT-FACE models are statistically indistinguishable from human annotators ($\Delta = 0.019$, $p = 0.103$), while significantly ($p < 0.001$) outperforming proprietary models, both multi-shot and zero-shot VLMs, and random baselines. These findings underscore that with focused dataset construction and careful fine-tuning, state-of-the-art models can genuinely approach human-level reliability on synthetic FER tasks. By contrast, general-purpose VLMs continue to lag behind, particularly when faced with nuanced or ambiguous facial expressions, highlighting persistent limitations despite recent advances.

To provide a more granular perspective, Figure 4 disaggregates results by individual model and prompt setup, offering insight into the underlying variability. Here, Spearman's $\rho$ measures rank

Table 3: Support for zero-shot and multi-shot emotion-analysis prompts (format only). *Gemini 2.5-pro-03-25 was discontinued after evaluating 1847 of 2500 images from EMONET-FACE HQ. ✓: successful parsing; ✗: refusal or unparseable output (e.g., mismatched labels)

| Model name | Zero-shot prompt | Multi-shot prompt | Comment |
|---|---|---|---|
| Gemini 2.5 Pro 03-25 | ✗ | ✓* | Empty Response |
| Gemini 2.5 Flash | ✓ | ✓ | - |
| Gemini 2.0 Flash | ✓ | ✓ | - |
| Claude 3.7 Sonnet | ✗ | ✓ | Empty Response |
| GPT-4o | ✓ | ✗ | Rejection |
| Nova Pro | ✓ | ✗ | Non-parsable Output |
| Pixtral Large | ✓ | ✗ | Non-parsable Output |
| Qwen Plus | ✓ | ✗ | Non-parsable Output |

correlation between model and human emotion ratings across all 40 categories. Our tailored models (EMPATHICINSIGHT-FACE) again match or exceed the performance of proprietary systems and VLMs, displaying consistency across runs. While a few VLMs (such as Gemini) achieve moderate agreement, many others fall well short, and there is no clear pattern favoring either zero-shot or multi-shot setups. This heterogeneity reinforces the "hit-or-miss" nature of current VLM annotators: depending on the model and prompt configuration, performance may range from reasonable to essentially random. As further illustrated in Table 3, prompt handling difficulties, model-internal safety constraints, and sensitivity to format further hinder VLM robustness—only two of nine VLMs consistently produce outputs under both prompt types. These results are also backed by direct comparison in App. Figure 6, which visually details the pairwise agreement landscape at the annotator level. This unpredictability highlights the need for careful selection and calibration when applying VLMs to FER tasks.

In summary, systematic evaluation of human, proprietary, and VLM-based annotators demonstrates that our new datasets and models reliably enable near-human performance, while exposing the current limitations of general-purpose AI models. EMONET-FACE thus establishes a rigorous new benchmark for progress in FER.

## 4.3   Generalization to Real-World Datasets

A key question for any benchmark based on synthetic data is how well models trained on it generalize to real-world, in-the-wild images. To assess this sim-to-real transfer, we evaluated our EMPATHICINSIGHT-FACE LARGE model on two established real-world datasets: the Facial Emotion Recognition Dataset (FERD; n=152) [41] and AffectNet (n=31,002) [33].

Before evaluation, a primary task is bridging the gap between our model's 40-category continuous output and the single discrete labels of the target datasets. To this end, we mapped our 40 labels to their 8-category taxonomies (details in App. D). After this mapping, we applied a softmax function across our model's 40 outputs to obtain a probability distribution. We then considered one fo the 8 basic emotion to be *present* if the probability of *any* of its constituent fine-grained emotions exceeded a threshold of $1.5 \times (1/40)$, which is 50% higher than a uniform random guess. This predicted label was then compared against the ground-truth label from the respective dataset to calculate accuracy.

As shown in Table 4, our model demonstrates strong generalization, achieving a mean accuracy of 78.29% on FERD and 75.72% on AffectNet. Performance is particularly high for categories like *Happy* and *Surprise*, where conceptual overlap in the category mapping is strong. Even with the inherent challenges of mapping between different emotion taxonomies, these results indicate that training on the controlled, diverse, and high-quality images of EMONET-FACE provides a robust foundation for real-world FER. This suggests that synthetic data can effectively bridge gaps in existing datasets, enabling the development of more capable and generalizable models.

## 5   Discussion

**Challenges and Chances in FER.**   FER remains an intrinsically complex and nuanced endeavor, rooted in fundamental questions about the nature of emotion itself. Decades of affective science, including the Theory of Constructed Emotion (TCE), emphasize that emotions are not universally

Table 4: Performance of our EMPATHICINSIGHT-FACE LARGE model on real-world emotion datasets (FERD and AffectNet). We report accuracy (%) after mapping our 40-category output to their 8 basic emotion categories using our threshold-based prediction strategy.

|          | anger | contempt | disgust | fear   | happy  | neutral | sad   | surprise | avg.  |
|----------|-------|----------|---------|--------|--------|---------|-------|----------|-------|
| FERD     | 73.68 | 31.58    | 78.95   | 100.00 | 100.00 | 84.21   | 78.95 | 78.95    | **78.29** |
| AffectNet| 77.05 | 28.75    | 40.53   | 69.08  | 99.25  | 78.96   | 83.94 | 98.70    | **75.72** |

"read out" from facial muscle configurations, but rather interpreted by observers through a dynamic interplay of visual cues, individual experience, and situational context (see App. A.1). Our analysis of inter-annotator agreement (Section 3) underscores this reality: even among domain experts, there is marked variability in how expressions are labeled, particularly for subtle categories. This diversity should not be seen as noise, but as a core finding. It reflects the genuine psychological complexity of emotion perception and demonstrates that EMONET-FACE captures this ambiguity. A benchmark with perfect agreement would be unrealistic and likely less robust. Our findings motivate a paradigm shift away from seeking a single authoritative label, toward approaches that estimate distributions over plausible emotions, a challenge for which EMONET-FACE provides a foundational testbed.

This subjectivity and inter-annotator diversity, should not be seen solely as sources of noise or error; rather, they highlight the deeply constructive and situationally-contingent character of affective perception. Our findings motivate a paradigm shift away from only seeking a single authoritative label for facial images, toward approaches that estimate distributions over plausible emotion categories (see Figure 5) or defer to LLMs and VLMs capable of subsequent context-sensitive reasoning.

In response to these challenges, our study introduces EMONET-FACE —a comprehensive suite designed to systematically benchmark both human and model performance. By providing a large-scale, diverse, and fine-grained test bed, EMONET-FACE enables robust examination of inter-annotator variability, the limits of facial expression alone for conveying emotion, and the reliability of both domain-specific and general-purpose AI in this domain. We hope this resource lays the groundwork for future research that not only seeks technical progress, but also more faithfully engages with the rich psychological and contextual complexity of human affect.

**Specialized Models Can Learn to See What Humans See.** Despite the inherent challenges of FER, our EMPATHICINSIGHT-FACE models exhibit consistently high agreement with human annotators. This success is partly attributable to our use of SigLIP2, a powerful vision backbone that provides strong image embeddings highly suited for affective analysis. As shown in Table 7 and Figure 6, our EMPATHICINSIGHT-FACE LARGE model achieves human agreement scores significantly above state-of-the-art zero-shot VLMs, proprietary systems, and even several individual expert annotators. This level of agreement is significant, given our annotators' strong domain expertise, demonstrating that targeted model design and careful data curation can substantially close the gap between machine and human performance on this task. Yet it is important to recognize that such agreement does not necessarily equate to genuine emotional understanding. Current models, including our own, may accurately mimic or imitate human emotion recognition on a surface level by leveraging statistical regularities in the data, without actually grasping the underlying emotional states or meaning. True comprehension of emotion likely requires not only pattern matching in facial cues, but also deeper insight into the context, causes, and subjective experience behind each expression.

**General-Purpose VLMs Hold Untapped Potential.** Our broader benchmarking of general-purpose VLMs reveals substantial inconsistencies across models and prompt setups: while a few VLMs, such as Gemini, achieve moderate agreement for certain settings, their performance is generally unreliable and highly sensitive to prompt formatting (see Figures 3 and 4). Crucially, there is no current prompt or model configuration that consistently enables VLMs to excel at nuanced FER tasks. Furthermore, we observe a pronounced misalignment problem: different VLMs (or even the same VLM under different prompts) frequently produce divergent or incoherent predictions for the same input image. This lack of alignment and robustness presents a serious challenge for deploying these systems in interactive, real-world settings, as human users require dependable and interpretable model behavior for meaningful interaction [15, 18]. Robustness and alignment—consistency within a model and agreement across models—are therefore just as important as absolute performance when

building trustworthy and user-friendly affective AI. While the foundational capabilities of VLMs are rapidly improving and may eventually surpass specialized approaches like our EMPATHICINSIGHT-FACE, realizing this potential will require addressing both their alignment and robustness shortcomings through new techniques, such as improved prompting strategies, architectural innovations, or alignment-focused training protocols. Future work bridging these gaps—potentially by combining the flexibility of VLMs with task-specific supervision—could unlock even greater advances, and enable robust, reliable, and context-aware emotion understanding at scale.

**Limitations.** Despite these advances, several limitations should be considered. *Sim-to-Real Generalization:* While our new experiments (Section 4.3) show promising generalization to real-world images, the sim-to-real gap remains an important area for future research. Validating performance across a wider range of naturalistic datasets is crucial. *Static vs. Dynamic Cues:* Our work is restricted to static facial expressions. This omits temporal dynamics, such as the evolution of an expression or microexpressions, which are vital for nuanced emotion interpretation. EmoNet-Face provides a foundation, but future work should extend this to dynamic video-based benchmarks. *Cross-Cultural Scope:* Our 40-category taxonomy, while comprehensive, is grounded in Western-centric psychological literature. Its universality across different cultures is an open question. Although we controlled for diverse ethnicities in image generation, dedicated cross-cultural validation is a critical next step. *Context-Free Analysis:* By focusing on faces in isolation, we remove contextual confounders but also limit the scope of analysis. As highlighted by TCE, context is integral to emotion perception. Integrating situational and multimodal cues (e.g., speech, body language) is essential for the next generation of emotionally intelligent AI.

In summary, the EMONET-FACE suite establishes a rigorous and versatile new testbed for emotion recognition research. Our results demonstrate that focused, expert-informed modeling can yield near-human—and in some cases, super-human—consistency on challenging FER tasks. Addressing the outlined limitations, particularly by embracing richer contextual and multimodal information, will be key to the next generation of emotionally intelligent AI systems.

**Ethical Considerations, Licensing, and Use.** This work addresses concerns about uncalibrated AI by providing a benchmark for safer, more nuanced emotion recognition. All images in EMONET-FACE are synthetic, mitigating privacy risks associated with human data. We acknowledge the risk of misuse (e.g., for manipulation) and release our work to support transparent, safety-oriented research.

To ensure clarity for downstream users, we specify our licensing. The EMONET-FACE datasets and EMPATHICINSIGHT-FACE models are released under the CC BY 4.0 license. This is permissible because the source T2I models grant us the necessary rights. For images generated with a paid Midjourneysubscription, the terms of service grant full ownership of the assets to the creator. For images generated with FLUX.1[dev], the license explicitly states that the creators claim no ownership of the outputs and they may be used for any purpose, including commercial. However, this permissive license does not override legal and regulatory obligations. Downstream applications must comply with local laws, such as the EU AI Act, which prohibits emotion inference in sensitive contexts like workplaces and educational institutions (Article 5(1)(f)). EMONET-FACE is not intended for such uses; its application should be confined to legally permitted and ethically sound research contexts.

# 6 Conclusion

We introduced the EMONET-FACE suite—a comprehensive resource featuring a fine-grained 40-category taxonomy, richly annotated datasets, and specialized models that establish new state-of-the-art benchmarks for FER. The EMONET-FACE benchmark exposes significant limitations in current tools and models for emotion evaluation, underscoring both the complexity of the task and the need for future innovation. Leveraging this suite, we developed the EMPATHICINSIGHT-FACE models, which deliver leading performance on facial emotion recognition while also illuminating the substantial variability in human judgments and the inherent constraints of using facial expressions alone. We provide a thorough discussion of these challenges and outline promising directions for advancing human-like emotion understanding in AI—impacting not only computer science, but also contributing valuable insights to psychology and the social sciences.

# 7 Acknowledgements

We gratefully acknowledge the support of Intel (oneAPI Center of Excellence), DFKI, Nous Research (providing cluster access and compute), TU Darmstadt, TIB–Leibniz Information Centre for Science and Technology, and Hessian.AI (providing compute and helpful discussions), and the open-source community for contributing to emotional AI. This work benefited from the ICT-48 Network of AI Research Excellence Center "TAILOR" (EU Horizon 2020, GA No 952215), the Hessian research priority program LOEWE within the project WhiteBox, the HMWK cluster projects "Adaptive Mind" and "Third Wave of AI", and from the NHR4CES. Furthermore, this work was partly funded by the Federal Ministry of Education and Research (BMBF) project "XEI" (FKZ 01IS24079B). This work has also benefited from the early stages of the cluster projects by the Deutsche Forschungsgemeinschaft (DFG, German Research Foundation) under Germany's Excellence Strategy—"Reasonable AI" (EXC-3057) and "The Adaptive Mind" (EXC-3066); funding will begin in 2026.

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

# A Appendices

## A.1 Theories of Emotion: From Universal Programs to Constructed Experiences.

Emotion theories span a spectrum from biologically grounded to constructivist accounts. Basic Emotion Theory (BET) posits a universal set of evolved emotions—anger, disgust, fear, happiness, sadness, and surprise—each with distinct physiological and facial expressions [13], though it has been critiqued for oversimplifying emotion and overlooking cultural variability [3]. Cognitive appraisal theories emphasize interpretation, with the Schachter-Singer model combining physiological arousal with contextual labeling [44], and Lazarus's framework highlighting evaluations of relevance and coping potential as key drivers of emotional response [26]. The Theory of Constructed Emotion (TCE) offers a constructivist view, arguing that emotions are not innate but are constructed through predictive brain mechanisms using culturally learned concepts [4]. TCE holds that emotional experiences emerge from interpreting interoceptive signals (valence and arousal) via culturally shaped categories, rejecting the idea of fixed "emotion circuits" and emphasizing variability across instances. This predictive, concept-driven view has major implications for AI, calling for models that interpret affect in context-sensitive, culturally informed ways rather than relying on static facial-emotion mappings. This is further visualized in Figure 5.

## A.2 Detailed Expert Annotator Recruitment and Process

To ensure the highest quality and precision of emotion labels in EMONET-FACE, we relied on expert annotation by psychology professionals, recruited through Upwork and selected for their verified academic degrees or substantial experience in the field. Our broad team of annotators, aged 18–44 and spanning diverse ethnic and linguistic backgrounds—e.g., Hispanic, European, South Asian, and Middle Eastern—with fluency in up to four languages, brought an inclusive perspective to the annotation process. All annotators (13 in total) participated voluntarily after giving informed consent and were fairly compensated at rates well above local standards, in keeping with the NeurIPS Code of Ethics. Annotators were instructed to follow our detailed, standardized guidelines, rather than relying solely on personal intuition, following [42, 43], promoting coherence and reproducibility. Open communication allowed for regular clarification of any ambiguities during annotation.

## A.3 EMONET-FACE Data Preprocessing

The preprocessing pipeline included: (1) Parsing Ratings—extracting structured emotion scores; (2) Emotion Extraction—identifying all unique emotion labels to define the target space; (3) Long Format Transformation—reshaping the dataset into (image, annotator, emotion) triplets, with missing ratings filled as 0 to reflect the annotation platform's assumption that unrated emotions indicate absence; and (4) Annotator Typing—labeling annotators as Human or VLM based on a binary flag, enabling separate analyses. All ratings were standardized on a 0–7 scale.

## A.4 Statistical Analysis and Visualization

The calculated agreement metrics were further analyzed and visualized:

- **Summary Statistics:** Mean, median, standard deviation, and count were calculated for Weighted Kappa and Spearman's Rho, grouped by the type of annotator pair (Human-Human, Human-Model, Model-Model). Similar summaries were computed for Krippendorff's Alpha, grouped by annotator group (Human, Model, All).

- **Visualizations:**
  - Box plots were generated to show the distribution of pairwise Weighted Kappa and Spearman's Rho across the different pair types.
  - A heatmap visualized the average Weighted Kappa for every annotator pair across all emotions. Annotators were ordered with human annotators listed first (alphabetically), followed by model annotators (alphabetically), for clarity.
  - Bar plots displayed Krippendorff's Alpha for the top N emotions (ranked by human annotator agreement) for the Human group, including reference lines indicating common interpretation thresholds (e.g., 0.80 for excellent, 0.67 for acceptable agreement).

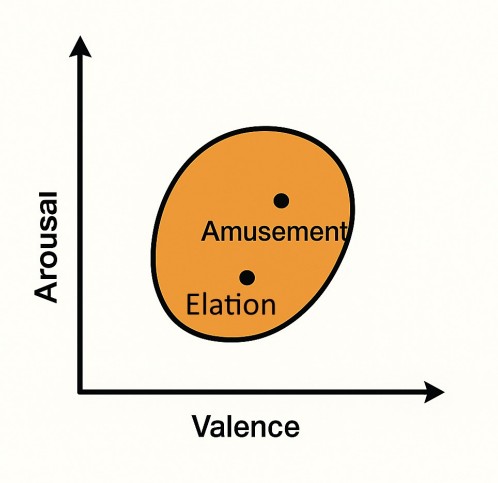

Figure 5: Rather than mapping each face to a single label, we estimate a distribution over plausible emotion categories.

- Bar plots showed the average pairwise Weighted Kappa per emotion, faceted by pair type.
- Statistical comparisons were performed using the Mann-Whitney U test to evaluate significant differences in Weighted Kappa distributions between the different pair types (e.g., Human-Human vs. Human-Model). The results, including the difference in means, bootstrapped 95% confidence intervals for the difference, and p-values, were annotated onto the Kappa distribution box plot using the 'statannotations' library.

- **Output Files:** Detailed results (pairwise scores, Alpha scores, summaries) were saved to CSV files in the 'output/tables' directory, and all generated plots were saved as PNG files in the 'output/figures' directory.

This comprehensive analysis allows for a detailed understanding of the reliability of the emotion ratings and the degree of alignment between human perception and VLM assessments within the EMO+ dataset.

The analysis was performed using Python (v3.10.12) with the libraries pandas (v2.2.2), numpy (v1.25.2), scikit-learn (v1.6.1), scipy, krippendorff (v0.8.1), seaborn (v0.13.2), matplotlib (v3.7.2), and statannotations (v0.7.2).

Table 5: The EmoNet-Face 40-Category Emotion Taxonomy. Each category represents a cluster of semantically related emotion words derived from the Handbook of Emotions and refined through expert consultation.

| Category Name | Associated Descriptive Words |
| --- | --- |
| 1. Amusement | 'lighthearted fun', 'amusement', 'mirth', 'joviality', 'laughter', 'playfulness', 'silliness', 'jesting' |
| 2. Elation | 'happiness', 'excitement', 'joy', 'exhilaration', 'delight', 'jubilation', 'bliss', 'Cheerfulness' |
| 3. Pleasure/Ecstasy | 'ecstasy', 'pleasure', 'bliss', 'rapture', 'Beatitude' |
| 4. Contentment | 'contentment', 'relaxation', 'peacefulness', 'calmness', 'satisfaction', 'Ease', 'Serenity', 'fulfillment', 'gladness', 'lightness', 'serenity', 'tranquility' |
| 5. Thankfulness/Gratitude | 'thankfulness', 'gratitude', 'appreciation', 'gratefulness' |
| 6. Affection | 'sympathy', 'compassion', 'warmth', 'trust', 'caring', 'Clemency', 'forgiveness', 'Devotion', 'Tenderness', 'Reverence' |

| Category Name | Associated Descriptive Words |
| --- | --- |
| 7. Infatuation | 'infatuation', 'having a crush', 'romantic desire', 'fondness', 'butterflies in the stomach', 'adoration' |
| 8. Hope/Optimism | 'hope', 'enthusiasm', 'optimism', 'Anticipation', 'Courage', 'Encouragement', 'Zeal', 'fervor', 'inspiration', 'Determination' |
| 9. Triumph | 'triumph', 'superiority' |
| 10. Pride | 'pride', 'dignity', 'self-confidently', 'honor', 'self-consciousness' |
| 11. Interest | 'interest', 'fascination', 'curiosity', 'intrigue' |
| 12. Awe | 'awe', 'awestruck', 'wonder' |
| 13. Astonishment/Surprise | 'astonishment', 'surprise', 'amazement', 'shock', 'startlement' |
| 14. Concentration | 'concentration', 'deep focus', 'engrossment', 'absorption', 'attention' |
| 15. Contemplation | 'contemplation', 'thoughtfulness', 'pondering', 'reflection', 'meditation', 'Brooding', 'Pensiveness' |
| 16. Relief | 'relief', 'respite', 'alleviation', 'solace', 'comfort', 'liberation' |
| 17. Longing | 'yearning', 'longing', 'pining', 'wistfulness', 'nostalgia', 'Craving', 'desire', 'Envy', 'homesickness', 'saudade' |
| 18. Teasing | 'teasing', 'bantering', 'mocking playfully', 'ribbing', 'provoking lightly' |
| 19. Impatience and Irritability | 'impatience', 'irritability', 'irritation', 'restlessness', 'short-temperedness', 'exasperation' |
| 20. Sexual Lust | 'sexual lust', 'carnal desire', 'lust', 'feeling horny', 'feeling turned on' |
| 21. Doubt | 'doubt', 'distrust', 'suspicion', 'skepticism', 'uncertainty', 'Pessimism' |
| 22. Fear | 'fear', 'terror', 'dread', 'apprehension', 'alarm', 'horror', 'panic', 'nervousness' |
| 23. Distress | 'worry', 'anxiety', 'unease', 'anguish', 'trepidation', 'Concern', 'Upset', 'pessimism', 'foreboding' |
| 24. Confusion | 'confusion', 'bewilderment', 'flabbergasted', 'disorientation', 'Perplexity' |
| 25. Embarrassment | 'embarrassment', 'shyness', 'mortification', 'discomfiture', 'awkwardness', 'Self-Consciousness' |
| 26. Shame | 'shame', 'guilt', 'remorse', 'humiliation', 'contrition' |
| 27. Disappointment | 'disappointment', 'regret', 'dismay', 'letdown', 'chagrin' |
| 28. Sadness | 'sadness', 'sorrow', 'grief', 'melancholy', 'Dejection', 'Despair', 'Self-Pity', 'Sullenness', 'heartache', 'mournfulness', 'misery' |
| 29. Bitterness | 'resentment', 'acrimony', 'bitterness', 'cynicism', 'rancor' |
| 30. Contempt | 'contempt', 'disapproval', 'scorn', 'disdain', 'loathing', 'Detestation' |
| 31. Disgust | 'disgust', 'revulsion', 'repulsion', 'abhorrence', 'loathing' |
| 32. Anger | 'anger', 'rage', 'fury', 'hate', 'irascibility', 'enragement', 'Vexation', 'Wrath', 'Peevishness', 'Annoyance' |
| 33. Malevolence/Malice | 'spite', 'sadism', 'malevolence', 'malice', 'desire to harm', 'schadenfreude' |
| 34. Sourness | 'sourness', 'tartness', 'acidity', 'acerbity', 'sharpness' |
| 35. Pain | 'physical pain', 'suffering', 'torment', 'ache', 'agony' |
| 36. Helplessness | 'helplessness', 'powerlessness', 'desperation', 'submission' |
| 37. Fatigue/Exhaustion | 'fatigue', 'exhaustion', 'weariness', 'lethargy', 'burnout', 'Weariness' |
| 38. Emotional Numbness | 'numbness', 'detachment', 'insensitivity', 'emotional blunting', 'apathy', 'existential void', 'boredom', 'stoicism', 'indifference' |
| 39. Intoxication/Altered States | 'being drunk', 'stupor', 'intoxication', 'disorientation', 'altered perception' |
| 40. Jealousy & Envy | 'jealousy', 'envy', 'covetousness' |

## A.5 Detailed Image Generation Models and Reproducibility

For transparency and reproducibility in the generation of the EMONET-FACE datasets, all prompt texts and specific model identifiers used for image creation will be released alongside the dataset.

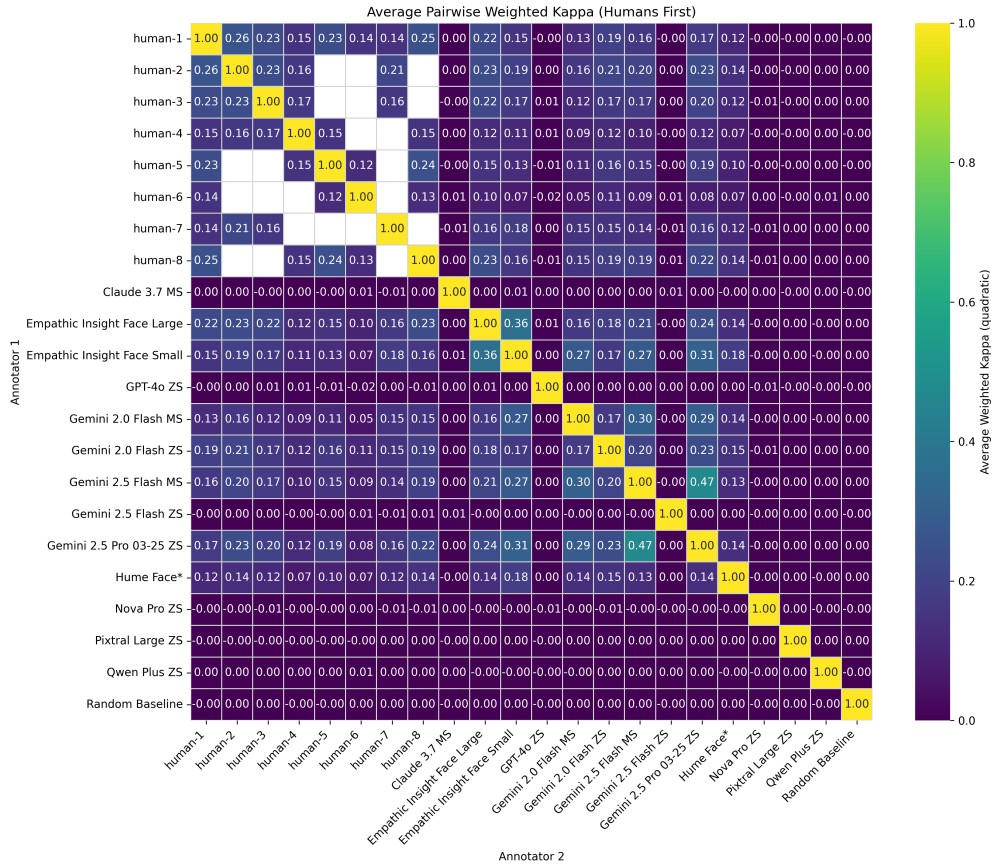

Figure 6: Heatmap of Average Pairwise Weighted Kappa ($\kappa_w$, quadratic weights) Across All Emotions. Annotators are grouped with humans listed first (alphabetically), followed by models (alphabetically). Values closer to 1 indicate higher agreement. The distinct block of warmer colors for Human-Human pairs highlights their higher relative agreement compared to Human-Model and Model-Model pairs.

Table 6: Internal Agreement for Repeated Annotations

| Annotator | Images Annotated | Internal Agreement |
|---|---|---|
| Annotator 1 | 2000 | 86.10% |
| Annotator 2 | 2000 | 96.20% |
| Annotator 3 | 2000 | 84.77% |

The EMONET-FACE HQ dataset (2,500 images) was generated using a combination of MidJourney v6 [32] and Flux-Dev and -Pro [6]. The larger EMONET-FACE BINARY (19,999 images) and EMONET-FACE BIG (over 203,000 images) datasets were produced exclusively with Flux-Dev.

We deliberately selected multiple state-of-the-art (SOTA) text-to-image (T2I) models to minimize potential confounders and ensure high image quality across the diverse range of emotions and demographic attributes. During this selection process, we intentionally excluded models such as Juggernaut-XL or Kolors from our final pipeline due to their propensity for frequent visual artifacts, including exaggerated wrinkles and unnatural facial features, which were not easily mitigated through prompt engineering. The prompts for all models specified sharp, close-up portraits with neutral backgrounds to maximize emotion clarity and consistently maintain demographic diversity across all generated datasets.

| Category | General Emotion | Example Emotions |
|---|---|---|
| **Positive High-Energy Emotions** | Amusement | lighthearted fun, amusement, mirth, joviality, laughter, playfulness, silliness, jesting, delight |
| | Elation | happiness, excitement, joy, exhilaration, laughter, jubilation, cheerfulness |
| | Pleasure/Ecstasy | ecstasy, pleasure, bliss, beatitude |
| | Hope/Enthusiasm/Optimism | hope, enthusiasm, optimism, anticipation, courage, encouragement, zeal, fervor, inspiration, determination |
| | Triumph | triumph, superiority, hubris |
| | Interest | interest, fascination, curiosity, need for cognition |
| | Awe | awe, awestruck, wonder |
| | Astonishment/Surprise | astonishment, surprise, amazement, shock, startlement |
| | Teasing | teasing, bantering, mocking playfully, ribbing, provoking lightly |
| **Positive Low-Energy Emotions** | Contentment | contentment, relaxation, peacefulness, calmness, satisfaction, ease, serenity, bliss, fulfillment, gladness, lightness, serenity, tranquility |
| | Thankfulness/Gratitude | thankfulness, gratitude, appreciation, gratefulness |
| | Affection | sympathy, compassion, warmth, trust, caring, forgiveness, devotion, tenderness, reverence |
| | Relief | relief, respite, alleviation, solace, comfort, liberation |
| | Contemplation | contemplation, thoughtfulness, pondering, reflection, meditation, brooding, pensiveness |
| | Pride | pride, dignity, self-confidently, honor, self-consciousness |
| **Negative High-Energy Emotions** | Fear | fear, terror, dread, apprehension, alarm, horror, panic, nervousness |
| | Anger | anger, rage, fury, hate, irascibility, enragement, wrath, annoyance |
| | Malevolence/Malice | spite, sadism, malevolence, malice, malicious envy, revenge, desire to harm, schadenfreude |
| | Disgust | disgust, revulsion, repulsion, abhorrence, loathing |
| | Impatience and Irritability | impatience, irritability, irritation, restlessness, short-temperedness, exasperation |
| | Distress | worry, anxiety, unease, anguish, trepidation, concern, upset, pessimism, foreboding |
| **Negative Low-Energy Emotions** | Sadness | sadness, sorrow, grief, melancholy, dejection, despair, self-pity, sullenness, heartache, mournfulness, misery |
| | Bitterness | resentment, acrimony, bitterness, cynicism, rancor, malicious envy |
| | Contempt | contempt, disapproval, scorn, disdain, loathing, detestation |
| | Disappointment | disappointment, regret, dismay, letdown, chagrin |
| | Shame | shame, guilt, remorse, humiliation, contrition |
| | Emotional Numbness | numbness, detachment, insensitivity, emotional blunting, apathy, existential void, boredom, stoicism, indifference |
| | Doubt | doubt, distrust, suspicion, skepticism, uncertainty, pessimism |
| | Jealousy & Envy | jealousy, envy, covetousness |
| | Embarrassment | embarrassment, shyness, mortification, discomfiture, awkwardness, self-consciousness |
| | Helplessness | helplessness, hopelessness, powerlessness, desperation, submission |
| **Cognitive States and Processes** | Concentration | concentration, deep focus, engrossment, absorption, attention |
| | Confusion | confusion, bewilderment, disorientation, perplexity |
| **Physical and Exhaustive States** | Pain | physical pain, suffering, torment, ache, agony |
| | Fatigue/Exhaustion | fatigue, exhaustion, weariness, lethargy, burnout, Weariness |
| | Intoxication/Altered States of Consciousness | being drunk, stupor, intoxication, disorientation, altered perception |
| | Sourness | sourness, tartness, acidity, acerbity, sharpness |
| **Longing & Lust** | Longing | yearning, longing, pining, wistfulness, nostalgia, craving, desire, benign envy, homesickness |
| | Sexual Lust | sexual lust, carnal desire, lust, feeling horny, feeling turned on |
| | Infatuation | infatuation, having a crush, romantic desire, fondness, butterflies in the stomach, adoration |

Figure 7: Emotion classification taxonomy showing hierarchical relationships between primary categories, general emotions, and specific descriptive terms. The complete classification (all levels) was used for `EmoNetHQ` annotations, while `EmoNetBinary` annotations utilized only the descriptive terms (bottom level).

# B  Dataset Construction Details

## B.1  Construction of EMONET-FACE BIG

The EMONET-FACE BIG dataset, designed primarily for large-scale model pre-training, was constructed through a multi-stage process. This involved an initial phase of broad-spectrum image generation and annotation, followed by targeted generation and a refined annotation strategy to ensure comprehensive coverage across our 40-emotion taxonomy.

**Initial Data Generation and Annotation**    The foundation of EMONET-FACE BIG was formed by incorporating all images from the EMONET-FACE BINARY dataset, augmented with an additional 20,000 images generated using the Flux text-to-image model. This combined set of images was then annotated using the Gemini 2.5 Flash model. The annotation prompt directed Gemini 2.5 Flash to identify the five most salient emotional dimensions present in each image and to assign an intensity score ranging from 0 to 7 for each of these selected dimensions. While this initial phase yielded a substantial volume of annotations, a qualitative review revealed that certain emotion categories within our taxonomy—such as infatuation, embarrassment, and sexual lust—received disproportionately few non-zero annotations from Gemini 2.5 Flash.

Table 7: Agreement summary (average weighted Kappa $\kappa_w$ with quadratic weights) for all annotators: mean, standard deviation, median, and top emotion (with highest agreement) per annotator. For humans, agreement is with all other humans; for models, agreement is with all humans. Table is sorted in descending order of the mean.

| Annotator | mean | std | median | top-1 |
|---|---|---|---|---|
| human-2 | 0.2185 | 0.1541 | 0.2010 | Elation (0.6330) |
| human-1 | 0.2012 | 0.1266 | 0.2139 | Elation (0.5259) |
| human-3 | 0.1972 | 0.1502 | 0.1832 | Elation (0.5766) |
| human-8 | 0.1908 | 0.1349 | 0.1562 | Elation (0.6365) |
| human-5 | 0.1857 | 0.1277 | 0.1646 | Elation (0.5866) |
| Empathic Insight Face Large | 0.1795 | 0.1217 | 0.2008 | Astonishment/Surprise (0.4981) |
| human-7 | 0.1720 | 0.1452 | 0.1546 | Anger (0.5689) |
| Gemini 2.5 Pro 03-25 ZS | 0.1707 | 0.1651 | 0.1192 | Elation (0.6025) |
| Gemini 2.0 Flash ZS | 0.1624 | 0.1562 | 0.1621 | Elation (0.6059) |
| human-4 | 0.1540 | 0.1448 | 0.1210 | Amusement (0.6290) |
| Gemini 2.5 Flash MS | 0.1502 | 0.1599 | 0.0855 | Elation (0.5824) |
| Empathic Insight Face Small | 0.1443 | 0.1715 | 0.0806 | Elation (0.6592) |
| human-6 | 0.1309 | 0.1057 | 0.1064 | Elation (0.4198) |
| Gemini 2.0 Flash MS | 0.1195 | 0.1522 | 0.0512 | Elation (0.5639) |
| Hume Face* | 0.1087 | 0.1429 | 0.0693 | Elation (0.6276) |
| Qwen Plus ZS | 0.0024 | 0.0047 | 0.0002 | Contentment (0.0167) |
| Gemini 2.5 Flash ZS | 0.0007 | 0.0085 | 0.0002 | Amusement (0.0219) |
| Random Baseline | 0.0003 | 0.0075 | 0.0015 | Elation (0.0243) |
| Claude 3.7 MS | -0.0002 | 0.0101 | -0.0000 | Pride (0.0198) |
| Pixtral Large ZS | -0.0004 | 0.0055 | -0.0008 | Amusement (0.0145) |
| GPT-4o ZS | -0.0022 | 0.0078 | -0.0007 | Anger (0.0108) |
| Nova Pro ZS | -0.0030 | 0.0100 | -0.0029 | Distress (0.0140) |

**Targeted Generation and Annotation with Hinting Strategy** To address the underrepresentation of these specific emotion categories and to enrich the dataset with more diverse examples for these less frequently annotated states, we implemented a revised two-step procedure:

1. **Targeted Image Generation:** We utilized the Flux-Dev model to generate new images. The prompts for Flux-Dev were specifically designed to elicit facial expressions corresponding to the underrepresented emotions. For instance, prompts targeted the generation of images depicting "embarrassment" or "infatuation."

2. **Annotation with a Hinting Strategy:** These newly generated, targeted images were subsequently annotated using Gemini Flash 2.0. The selection of Gemini Flash 2.0 for this phase was partly influenced by more favorable API rate limits available at the time of data collection, which was crucial for the iterative process. The core annotation instruction remained to score the top five most evident emotional dimensions on the 0-7 scale. However, a key modification was introduced: a "hint" was provided to the model. For example, if an image was generated with "infatuation" as the target emotion in the prompt, the annotation prompt for Gemini Flash 2.0 would include a suggestion that the image *might* contain "infatuation." This was intended to encourage the model to consider this specific dimension more readily, without mandating its selection or biasing the intensity score.

This hinting strategy proved effective. Qualitative inspection of the results indicated that for approximately 50% of the images where a hint was provided for a specific target emotion, Gemini Flash 2.0 assigned a non-zero annotation to that particular dimension. The quality of these hinted annotations was deemed reasonable and sufficient for the purposes of large-scale pre-training.

**Iterative Refinement and Final Dataset Composition** We iteratively applied this targeted image generation (with Flux-Dev) and hinted annotation (with Gemini Flash 2.0) process. The primary objective of this iterative refinement was to ensure that each of the 40 emotional dimensions in our taxonomy received approximately 5,000 samples with a non-zero annotation score attributed by Gemini Flash 2.0 (under the described hinting strategy). This iterative process continued until the desired representation was achieved for all emotion categories.

The final EMONET-FACE BIG dataset comprises a total of 203,201 images, each with synthetic emotion annotations derived through this combined approach of broad-spectrum initial generation/annotation and subsequent targeted generation/hinted-annotation.

## B.2    Training of Baseline Models

To establish baseline performance for emotion dimension prediction, we developed and trained several Multi-Layer Perceptron (MLP) models. The process began with feature extraction: we computed SIGLIP2-400M image embeddings (1152 dimensions) for all images within the EMoNet-Face Big, and EmoNetFace Binary datasets. These embeddings served as the input features for our models.

Our initial approach involved training 40 distinct MLP models, one for each of the 40 emotion dimensions. Each MLP was designed to take an image's SIGLIP2-400M embedding as input and predict the rating for its specific emotion dimension as a continuous score (ranging from 0 to 7). To calibrate these continuous predictions, we addressed potential biases in the models' interpretation of neutral expressions. For each of the 40 MLP models, we performed inference on 1700 images generated by FluxDev. These images depicted faces with neutral emotional expressions, created using prompts randomly varied across the complete template distribution for different ages, ethnic backgrounds, and genders. We then calculated the average rating assigned by each MLP model to these neutral faces. This average neutral rating, which varied per model, was subsequently subtracted from all predictions made by that specific MLP to correct for its baseline offset on neutral expressions.

We first developed a "small model" configuration. For these small models, training was conducted exclusively on the EMoNet-Face Big dataset. To address class imbalance inherent in the 0-7 rating scale for each emotion dimension, we employed a specific sampling strategy, referred to as the "stumbling strategy," to construct balanced training datasets. For a given emotion dimension, samples were first grouped into eight buckets corresponding to ratings 0 through 7. We then calculated the average number of images across these eight buckets. The training set for that dimension was constructed by sampling up to 25 percent of this average count from each bucket. This approach was adopted because the bucket for rating '0' (emotion not present) typically contained a significantly larger number of samples (e.g., 180,000-190,000) compared to other rating buckets. Consequently, the average number of samples per bucket might be around 20,000, leading to approximately 5,000 samples from the '0' bucket and proportionally fewer (e.g., around 1,000) from each of the other buckets (ratings 1-7). A validation set was created by reserving 5 percent of the samples from each bucket using the same principle. The small MLP architecture consisted of an input layer (1152 features), a hidden layer with 128 neurons (ReLU, Dropout 0.1), a second hidden layer with 32 neurons (ReLU, Dropout 0.1), and an output layer with 1 neuron. This architecture has 151,745 parameters. The checkpoint performing best on the validation set was saved.

Following experiments with the small model, a grid search over different model sizes indicated that a larger MLP architecture could yield modest improvements in validation accuracy. This led us to transition our experiments to a "big model." The big MLP architecture is defined as follows: an input layer (1152 features); a first hidden layer with 1024 neurons (ReLU, Dropout 0.2); a second hidden layer with 512 neurons (ReLU, Dropout 0.2); a third hidden layer with 256 neurons (ReLU, Dropout 0.2); and an output layer with 1 neuron. This larger architecture comprises 1,837,057 parameters.

Initially, this big model was trained on EMoNet-Face Big using the same strategy of balancing datasets for each emotion dimension by sampling from each 0-7 rating bucket up to a limit (25 percent of the average samples per bucket) to reduce the dominance of the overrepresented zero-rating for continuous score prediction as the small model. However, after these initial explorations, we shifted the dataset composition and prediction task. For each emotion dimension, we transformed the problem into a binary classification task. Images with a rating of 0 were assigned to a '0' bucket (emotion not present). All images with ratings from 1 to 7 were merged into a single '1' bucket (emotion present, irrespective of intensity). The big model was then trained to predict either 0 or 1 via regression. The EMoNet-Face Big dataset was prepared for this binary task by binarizing its ratings as described. Due to the '0' bucket being substantially larger than the '1' bucket even after binarization, we balanced the EMoNet-Face Big training data by downsampling the '0' bucket to match the number of samples in the '1' bucket. The human-annotated EmoNetFace Binary dataset was also prepared by binarizing its ratings in the same manner.

We explored different training regimes for the big binary model. This included training solely on the (binarized) human-annotated EmoNetFace Binary dataset and training solely on the (binarized and balanced) synthetic EMoNet-Face Big dataset. Through experiments on a validation set (sampled as 5 percent from each bucket of the respective training data), we determined that the optimal performance was achieved by first pre-training the model on the large, synthetically annotated EMoNet-Face Big dataset and subsequently fine-tuning it on the smaller, human-annotated EmoNetFace Binary dataset.

Our final big model was trained using this two-stage approach. First, it was pre-trained for 5 epochs on the binarized and balanced EMoNet-Face Big dataset. For this stage, we used a learning rate of 5e-5 and a ReduceLROnPlateau learning rate scheduler (monitoring validation loss, factor=0.2, patience=10). After pre-training, the training was restarted with the same initial learning rate of 5e-5 on the binarized human-annotated EmoNetFace Binary dataset. Fine-tuning on this dataset continued for 200 epochs. Throughout both training stages, the checkpoint that achieved the best performance on the corresponding validation set was saved for later use.

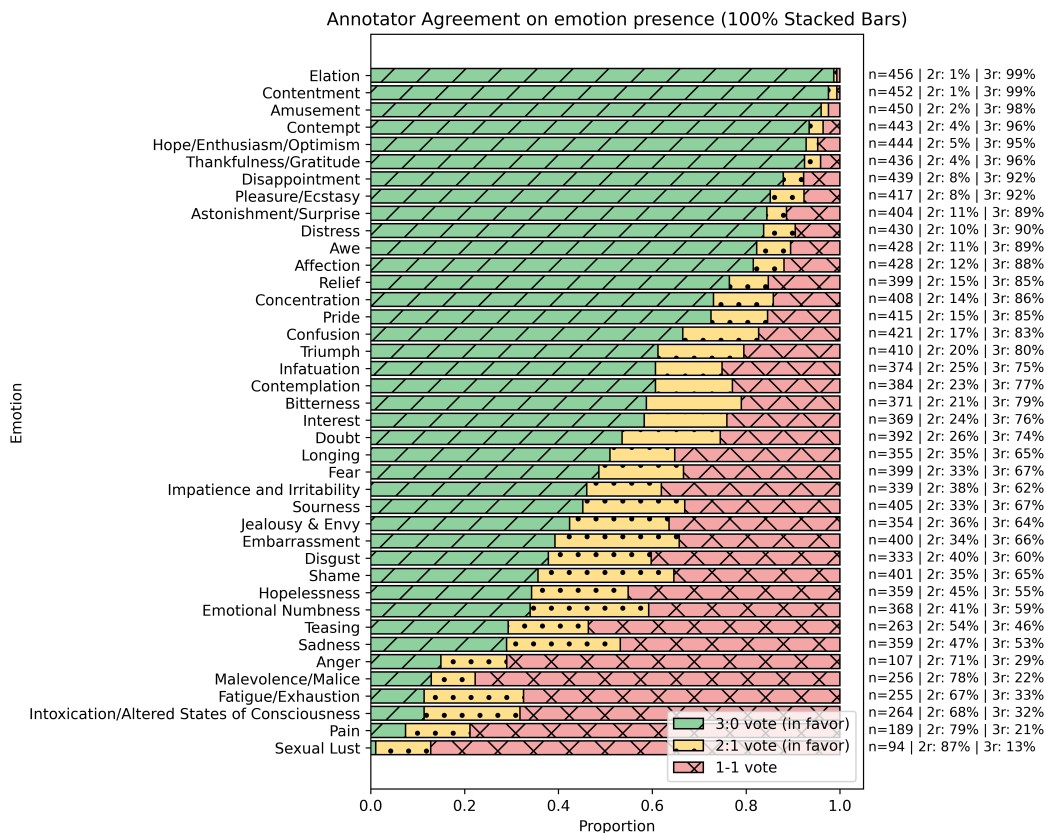

Figure 8: Annotator agreement for Human Annotators (n=4) in EMONET-FACE BINARY. Stacked bars show the proportion of image-emotion pairs with full agreement (all annotators in favor of emotion presence), partial agreement (2:1 split in favor), and disagreement (1:1 split) for each emotion. The numbers to the right of each bar indicate the total number of samples ($n$) and the percentage of cases rated by 2 or 3 annotators. Note that our annotation process was designed to only annotate positively rated image a second time, and double-positively rated images a third time.

## B.3 Preference Study: Model vs. Human Perception Alignment

To evaluate how well our model (`Empathic Insight Face Large`) aligns with human perception, we conducted a blind preference study using 999 of 2,500 total images of EMONET-FACE HQ (see Appendix C for UI details). Two annotators, not involved in initial labeling, each reviewed 499 to 500 images and compared two anonymized emotion label sets: (1) the human median of the five highest-rated emotions and (2) the model's five top-confidence predictions. Annotators selected the set that best matched the depicted expression. Of 999 collected judgments, 707 (70.77%)

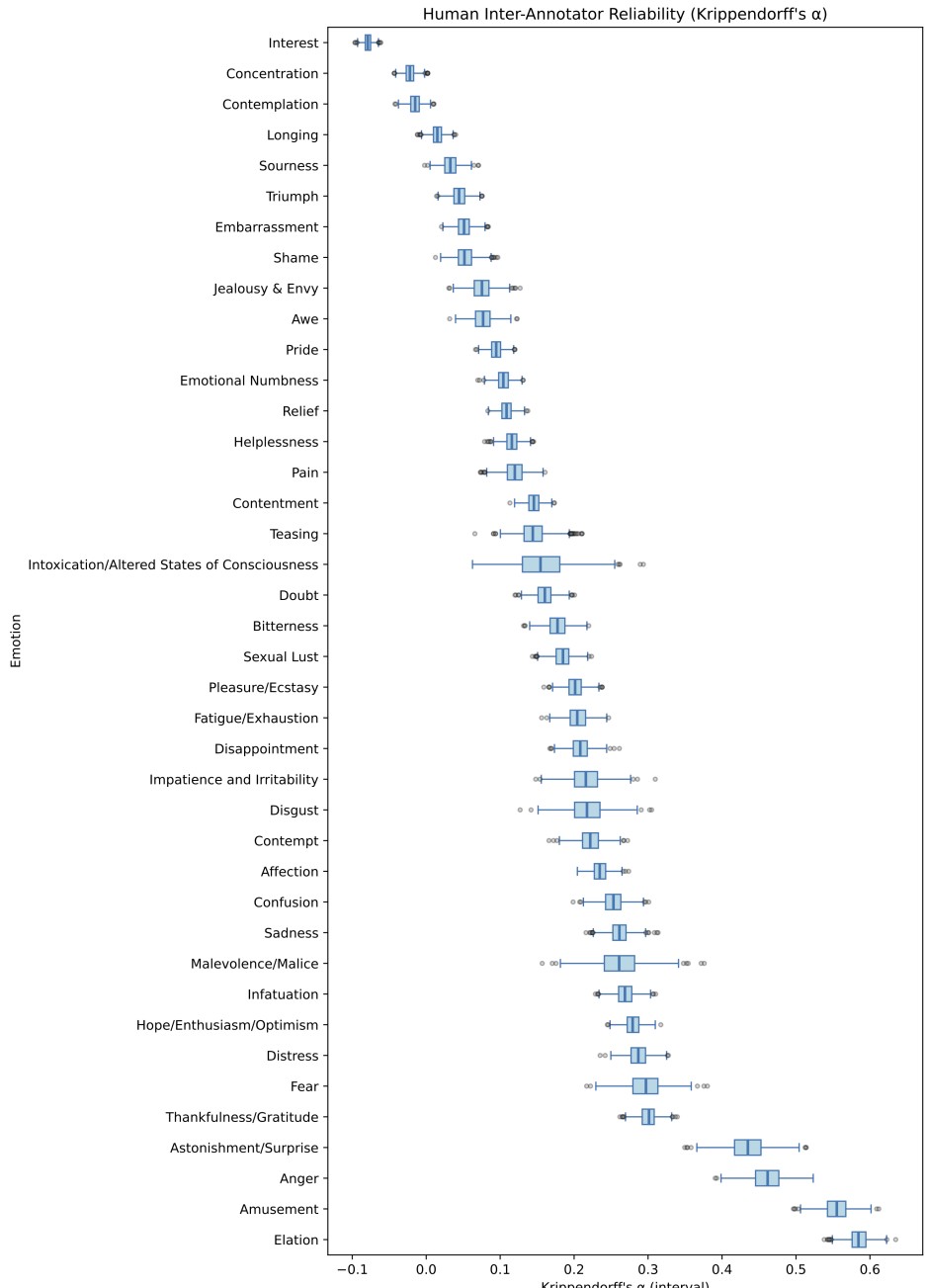

Figure 9: Krippendorff's Alpha ($\alpha$, interval level) for Human Annotators (n=8) in EMONET-FACE HQ. Boxplot depicts median, IQR, and outliers across all 40 Emotions, with higher bars indicate better agreement.

favored human labels, while 292 (29.23%) preferred the model's output. Though human experts were preferred overall, our model's top 5 predictions had been preferred to the average of 4 human expert annotations in almost 30% of the samples. These results suggest the model can remarkably well approximate human judgment, especially in contexts where human annotation is impractical due to scale or cost. Future work may focus on fine-tuning to address specific expression types where model performance lags.


| emotion | human-1 | human-2 | human-3 | human-7 | gpt-4o | nova-pro | pixtral-large-latest | qwen-plus |
|---|---|---|---|---|---|---|---|---|
| Cognitive States and Processes Concentration | 5 | 2 | 0 | 0 | 5 | 5 | 4 | 4 |
| Cognitive States and Processes Confusion | 0 | 0 | 0 | 0 | 4 | 3 | 2 | 3 |
| Longing & Lust Infatuation | 4 | 0 | 0 | 0 | 0 | 0 | 0 | 0 |
| Longing & Lust Sexual Lust | 1 | 0 | 0 | 0 | 0 | 0 | 0 | 0 |
| Negative High-Energy Emotions Anger | 0 | 0 | 0 | 0 | 2 | 6 | 1 | 1 |
| Negative High-Energy Emotions Disgust | 0 | 0 | 0 | 0 | 2 | 3 | 2 | 1 |
| Negative High-Energy Emotions Distress | 0 | 0 | 0 | 0 | 3 | 3 | 3 | 3 |
| Negative High-Energy Emotions Fear | 0 | 0 | 0 | 0 | 1 | 4 | 2 | 1 |
| Negative High-Energy Emotions Impatience and Irritability | 0 | 0 | 0 | 0 | 1 | 3 | 2 | 2 |
| Negative High-Energy Emotions Malevolence/Malice | 0 | 0 | 0 | 0 | 3 | 2 | 1 | 1 |
| Negative Low-Energy Emotions Bitterness | 0 | 0 | 0 | 0 | 1 | 2 | 2 | 1 |
| Negative Low-Energy Emotions Contempt | 0 | 0 | 0 | 0 | 1 | 3 | 1 | 1 |
| Negative Low-Energy Emotions Disappointment | 0 | 0 | 0 | 0 | 2 | 4 | 3 | 2 |
| Negative Low-Energy Emotions Doubt | 0 | 0 | 0 | 0 | 2 | 3 | 3 | 2 |
| Negative Low-Energy Emotions Emotional Numbness | 0 | 0 | 0 | 0 | 3 | 1 | 1 | 1 |
| Negative Low-Energy Emotions Jealousy & Envy | 0 | 0 | 0 | 0 | 1 | 3 | 2 | 1 |
| Negative Low-Energy Emotions Sadness | 0 | 0 | 0 | 0 | 4 | 4 | 4 | 1 |
| Negative Low-Energy Emotions Shame | 0 | 0 | 0 | 0 | 1 | 2 | 2 | 1 |
| Physical and Exhaustive States Fatigue/Exhaustion | 0 | 0 | 0 | 0 | 2 | 2 | 3 | 2 |
| Physical and Exhaustive States Intoxication/Altered States of Consciousness | 0 | 0 | 0 | 0 | 1 | 1 | 1 | 1 |
| Positive High-Energy Emotions Amusement | 2 | 3 | 0 | 0 | 0 | 0 | 0 | 0 |
| Positive High-Energy Emotions Astonishment/Surprise | 0 | 0 | 0 | 0 | 5 | 5 | 4 | 6 |
| Positive High-Energy Emotions Awe | 0 | 0 | 0 | 0 | 4 | 2 | 1 | 4 |
| Positive High-Energy Emotions Elation | 4 | 0 | 3 | 2 | 0 | 0 | 0 | 0 |
| Positive High-Energy Emotions Hope/Enthusiasm/Optimism | 4 | 0 | 4 | 0 | 0 | 0 | 0 | 0 |
| Positive High-Energy Emotions Interest | 5 | 2 | 0 | 0 | 2 | 6 | 3 | 5 |
| Positive High-Energy Emotions Pleasure/Ecstasy | 1 | 1 | 4 | 0 | 0 | 0 | 0 | 0 |
| Positive High-Energy Emotions Teasing | 4 | 0 | 0 | 0 | 0 | 0 | 0 | 0 |
| Positive High-Energy Emotions Triumph | 2 | 3 | 0 | 3 | 3 | 2 | 2 | 3 |
| Positive Low-Energy Emotions Affection | 5 | 2 | 5 | 0 | 0 | 0 | 0 | 0 |
| Positive Low-Energy Emotions Contemplation | 5 | 3 | 0 | 0 | 1 | 4 | 5 | 7 |
| Positive Low-Energy Emotions Contentment | 4 | 2 | 5 | 2 | 5 | 1 | 3 | 3 |
| Positive Low-Energy Emotions Pride | 3 | 4 | 7 | 4 | 4 | 5 | 2 | 2 |
| Positive Low-Energy Emotions Relief | 5 | 0 | 5 | 0 | 0 | 0 | 0 | 0 |
| Positive Low-Energy Emotions Thankfulness/Gratitude | 5 | 0 | 6 | 0 | 0 | 0 | 0 | 0 |

Figure 10: Discrepancies between human annotators and zero-shot prompted VLMs.

## C   Annotation Platform Instructions and UI

# Instructions

This project is about assessing the emotions that a person, shown in an image, appears to be feeling, according to **your opinion**. All faces that you find in this project are generated by AI and do not resemble any real existing person. This study aims to improvethe abilities of AI assistants to empathetically react to their user's feelings in contexts like counseling, therapy, and social support.

To help these systems to make a realistic assessment of their users' feelings, it is important that you fulfill these tasks with your honest **uncensored** and **spontaneous intuitions**.

Do not hesitate to omit certain categories you consider irrelevant for a given image or rate other categories highly if you feel this is appropriate.

Emotion recognition is always influenced by cultural and social factors and is therefore never 100% objective. Therefore, **don't worry too much about whether your evaluation is right or wrong**, but listen to your intuition and first impressions. After chosing the categories, you will adjust the slider from 0 (emotions not present) over 1 (emotions are weakly present) to 7 (the emotions are very strong). If you are unsure, take another look at the face and ask yourself how much this person seems to be feeling for the given emotions right now, and then adjust the slider to the number that comes to your mind.

Whatever comes to your mind: **It is okay**.

**Recommendations**
  1) Close your eyes for a moment when you start annotating a new sample and focus for ~10 seconds on your breath, until you feel calm & relaxed. Feeling relaxed will make you more perceptive of small nuances in the face and make it easier for you to listen to your intuition.
  2) Then look at the face & ask yourself for each category: **How much does the category seem to describe what this person seems to be feeling**?
  3) Repeat the process for the emotions: **How much do the given emotions seem to describe what this person seems to be feeling**? Instead of if a category is present or not (clicked or not), you now adjust the slider accordingly (leave it as it is for not present, or slide it on a value between 1 to 7 if present). Trust your gut feeling, your subjective intuition. It is fine.
  4) After you have processed all emotions move to the next sample starting with step 1) by clicking  Submit and Proceed .

Do the annotations in a calm environment in a relaxed state of mind. A tea or coffee (with or without cookies) and listening to relaxing music might help.

Listen to your heart, **trust your intuition & try to have fun**. 🙂

Figure 11: Instructions given to the human annotator for the expert annotation of EMONET-FACE HQ.

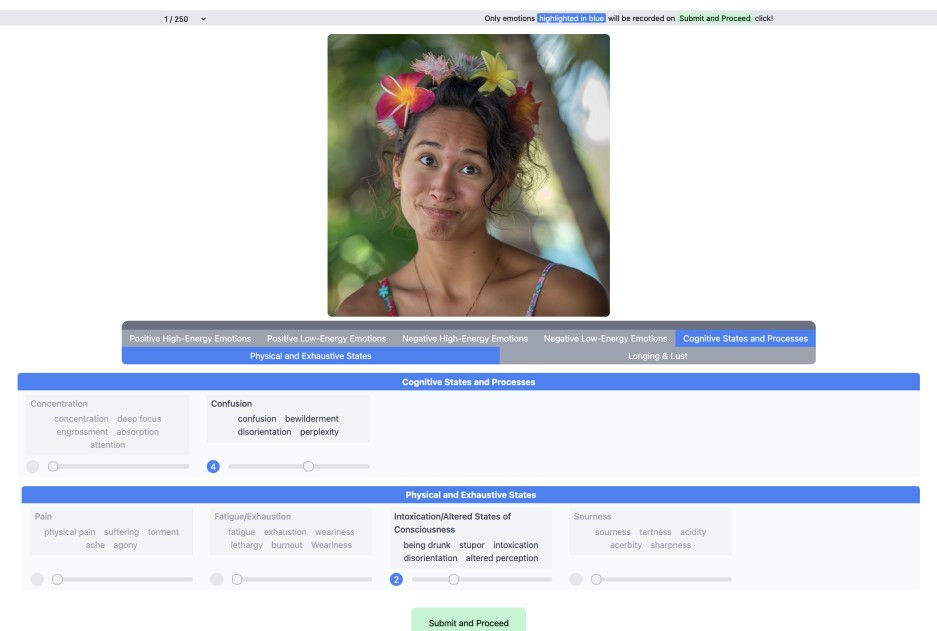

Figure 12: UI of our expert annotation tool for EMONET-FACE HQ.

## D    Taxonomy Mapping for Real-World Generalization

To assess the generalization capability of our EMPATHICINSIGHT-FACE model on established real-world emotion recognition datasets such as FERD and AffectNet (see Section 4.3), we mapped our fine-grained 40-category EMONET-FACE taxonomy to the 8 basic emotion categories used in these benchmarks. Because FERD and AffectNet differ slightly in their label definitions, we derived separate mappings for each dataset, as detailed in Table 8. This mapping process necessarily involves some loss of granularity, as nuanced affective states are compressed into broader emotion classes. For instance, both *Elation* and *Amusement* are mapped to *Happy*, while *Awe* corresponds to *Surprise* on FERD but to *Fear* on AffectNet. The resulting target categories—*Anger*, *Contempt*, *Disgust*, *Fear*, *Happy*, *Neutral*, *Sad*, and *Surprise*—align with the shared emotion taxonomy across these benchmarks.

# Emotion Classification Tutorial

## Instructions

In this task, you'll be assessing whether a specific emotion appears to be present in a face. Each face you see is generated by AI and does not resemble any real person. The goal of this study is to improve AI's ability to empathize by recognizing emotions in counseling, therapy, and social support settings.

To help make these systems better at detecting emotions, your honest, uncensored, and spontaneous responses are essential.

## Task Overview

You will be shown a face along with a list of emotions. For each emotion, your task is simply to decide Yes (emotion is present) or No (emotion is not present). This is not a precise science, so rely on your intuition and first impressions to make your decisions.

Remember, emotion recognition is influenced by cultural and social factors and is therefore subjective. There are no "right" or "wrong" answers here — just your honest responses based on how you perceive the expressions in the faces.

## Recommendations

- **Relax First:** Close your eyes for a moment before starting a new image and take a few deep breaths. This can help you feel calm and improve your awareness of subtle facial cues.

- **Observe & Decide:** Look at the face and consider each emotion in turn. Ask yourself, "Does this person seem to be feeling this emotion right now?" Then answer Yes or No based on your gut feeling.

- **Repeat & Submit:** After answering for each emotion, click Submit and Proceed to move to the next image. Repeat the process with the same relaxed focus.

- **Create a Calm Environment:** Working in a calm space with minimal distractions, possibly with a tea or coffee, can make the task more enjoyable and enhance your intuition.

Figure 13: Instructions given to the human annotator for the expert annotation of EMONET-FACE BINARY.

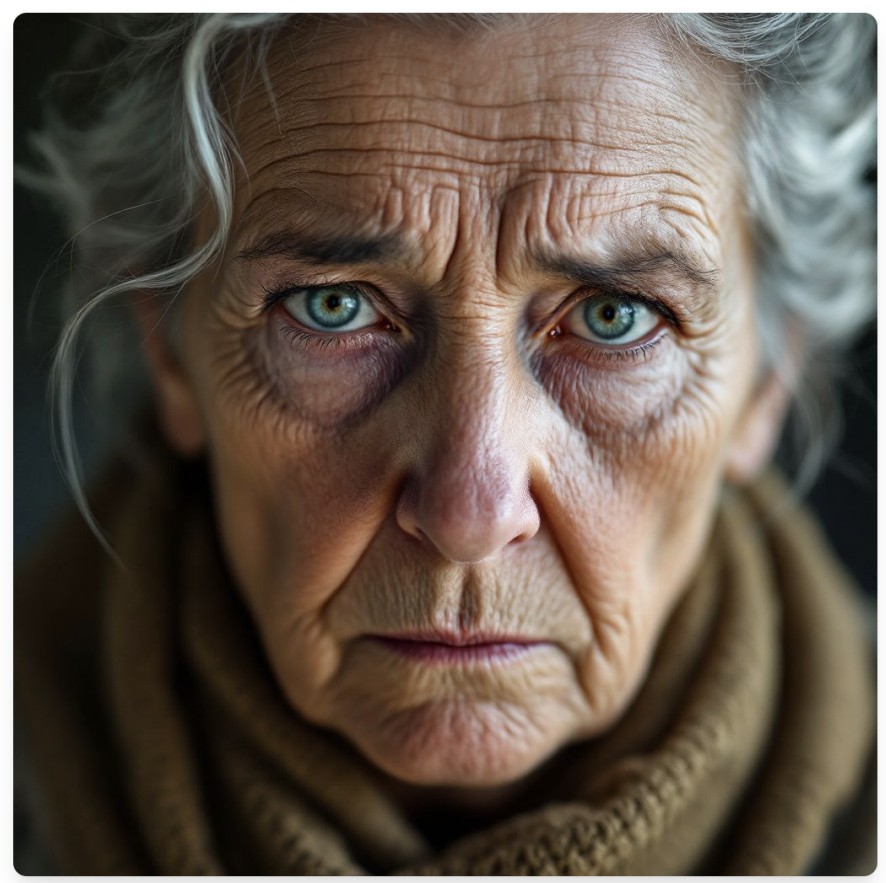

resentment, acrimony, bitterness, cynicism, rancor

Are the emotions present in the image?
*(You can also click "y" for Yes and "n" for No on the keyboard.)*

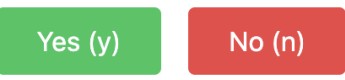

Progress: 2 / 2000 | Last annotation: 57486ms

Figure 14: UI of the self-developed expert annotation tool for EMONET-FACE BINARY.

## Preference Annotation Task

**Please follow these steps for the following annotation task**

1. Examine the person in the image carefully.

2. Choose the emotion group (left or right) that most accurately reflects the person's emotions and facial expression. Each group displays up to the top five emotions, ranked in descending order by annotator or model score.

3. Submit your choice by either:
   - Pressing the ← or → arrow key
   - Clicking the corresponding on-screen arrow button

*Please work systematically and trust your first impression. Thank you for your careful annotations.*

Figure 15: Instructions given to the human annotators for the preference annotation of of EMONET-FACE HQ.

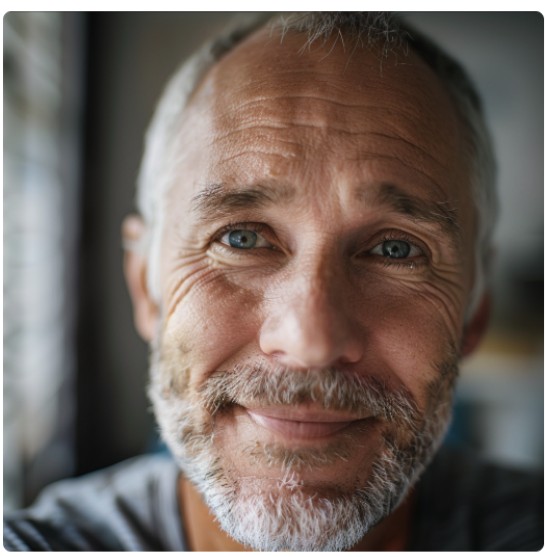

Use the **left** and **right** arrow keys, or click the **arrow buttons**, to choose the emotions that best describe the person.

| Elation | Contemplation |
|---|---|
| Amusement | Contentment |
| Contentment | Concentration |
| Thankfulness/Gratitude | Affection |
| Hope/Enthusiasm/Optimism | Thankfulness/Gratitude |

Figure 16: UI of our expert preference annotation tool for EMONET-FACE HQ.

Table 8: Mapping from our 40-category EMONET-FACE taxonomy to the 8 basic emotion categories used for evaluation on the FERD and AffectNet datasets.

| EMONET-FACE Category | FERD Mapping | AffectNet Mapping |
|---|---|---|
| Affection | Happy | Neutral |
| Anger | Anger | Anger |
| Astonishment / Surprise | Fear | Surprise |
| Awe | Surprise | Fear |
| Bitterness | Sad | Sad |
| Concentration | Neutral | Neutral |
| Contemplation | Neutral | Neutral |
| Contempt | Contempt | Contempt |
| Contentment | Neutral | Neutral |
| Confusion | Surprise | Neutral |
| Disappointment | Sad | Anger |
| Disgust | Disgust | Disgust |
| Distress | Fear | Sad |
| Doubt | Fear | Fear |
| Elation | Happy | Happy |
| Embarrassment | Anger | Anger |
| Emotional Numbness | Neutral | Neutral |
| Fatigue / Exhaustion | Neutral | Neutral |
| Fear | Surprise | Fear |
| Helplessness | Fear | Sad |
| Hope / Enthusiasm / Optimism | Happy | Neutral |
| Impatience and Irritability | Anger | Anger |
| Infatuation | Happy | Sad |
| Interest | Neutral | Neutral |
| Intoxication / Altered States of Consciousness | Neutral | Neutral |
| Jealousy & Envy | Anger | Disgust |
| Lighthearted Fun / Amusement | Happy | Happy |
| Longing / Yearning | Sad | Sad |
| Malevolence / Malice | Contempt | Anger |
| Pain | Sad | Sad |
| Pleasure / Ecstasy | Surprise | Surprise |
| Pride | Happy | Happy |
| Relief | Neutral | Neutral |
| Sadness | Sad | Sad |
| Sexual Lust | Neutral | Neutral |
| Shame | Sad | Disgust |
| Sourness | Disgust | Disgust |
| Teasing | Contempt | Contempt |
| Thankfulness / Gratitude | Happy | Neutral |
| Triumph | Happy | Surprise |

