# OpenReview forum: "EmoNet-Face: An Expert-Annotated Benchmark for Synthetic Emotion Recognition"
_NeurIPS.cc/2025/Datasets_and_Benchmarks_Track — NeurIPS 2025 Datasets and Benchmarks Track poster_

### Official Review · Reviewer_WZ4Z · 2025-06-30

**Ethics Flags:** Data quality and representativeness
**Rating:** 4
**Confidence:** 4

**Summary:**

This paper presents EmoNet-Face—an expert-annotated benchmark suite for synthetic facial expression recognition, comprising: (1) A 40-category fine-grained emotion taxonomy grounded in psychological theory; (2) Three large-scale AI-generated datasets (HQ/BINARY/BIG) with controlled demographic diversity; (3) Multi-expert annotation protocols with quality analysis; and (4) The EMPATHICINSIGHT-FACE model achieving human-expert-level performance.

**Dataset Code Accessibility:**

Yes

**Dataset Code Comments:**

The paper provides an open-source link.

**Ethical Comments:**

No significant ethical concerns; synthetic faces mitigate privacy risks.

**Ethical Considerations:**

Yes, there are ethics concerns that require attention by the authors

**Final Justification:**

The rebuttal satisfactorily addresses my technical concerns. However, considering the standards of a top-tier conference, I recommend borderline accept for this synthetic dataset.

**Limitations Weaknesses:**

1. The dataset is generated using different SOTA T2I models. The quality filtering of the data produced by these models was performed manually, which is overly time-consuming and labor-intensive.
2. If the model returns correctly formatted and parsable emotion labels, the output is considered successful. However, the paper does not seem to provide the success rate of the baseline model's outputs, and relevance is calculated only for successful outputs. If there are many incorrectly formatted outputs, but those with correct formats are semantically accurate, the results might be inflated. It is recommended to include the success rate of model outputs as an additional metric.
3. The proposed benchmark model was only tested on the EmoNet-Face dataset and not validated on other synthetic datasets, raising concerns about the method's generalizability.
4. The proposed facial expression dataset is a static image-based dataset rather than a dynamic video-based one, meaning it cannot capture the temporal evolution of expressions, especially microexpressions.

**Strengths Contributions:**

1. A fine-grained emotion dataset, EmoNet-Face, was established. A fine-grained classification encompassing 40 emotion categories such as "jealousy" and "exhaustion" was proposed. Based on terminology clustering from the "Handbook of Emotions" and expert validation, it supports multi-label annotation to accommodate emotional ambiguity.
2. A benchmark for facial expression recognition on the EmoNet-Face dataset, named EMPATHICINSIGHT-FACE, was proposed. Compared to other state-of-the-art (SOTA) models, this model achieved higher consistency with human annotators.

---

> ### Author Rebuttal · Authors · 2025-07-31
>
> We thank Reviewer WZ4Z for their feedback and for acknowledging our fine-grained taxonomy and high-performing baseline model. Below, we have addressed the points raised:
>
> **W1 Manual Filtering Being Labor-Intensive**: We agree that manual filtering is labor-intensive. Yet, we respectfully disagree that this thorough quality assurance is a weakness. Manual expert curation was essential to create a gold-standard benchmark - automated filtering cannot reliably detect subtle artifacts or ambiguous expressions that would compromise fine-grained emotion recognition. This rigorous approach distinguishes EmoNet-Face from noisier, automatically-collected datasets and ensures its value as a reliable evaluation resource. Moreover, and even in contrast, models trained on this high-quality data can now perform emotion annotation at scale, significantly reducing the need for manual labor in future applications and data. We have clarified this more prominently in the paper.
>
> **W2 Success Rate of VLM Outputs**: We did report this metric in Table 3, which explicitly details success and failure rates for all evaluated VLMs under both zero-shot and multi-shot prompting. The high failure rates (refusals, unparseable outputs) of many state-of-the-art VLMs are actually a significant finding of our benchmark, highlighting their brittleness for this task. Your point about combined metrics is valuable. Based on your suggestion, we have now computed another score, assuming all incorrectly formatted outputs are also semantically wrong. In this case, the actual performance of those models is even lower, further demonstrating the need for specialized models like EmpathicInsight-Face. We have adjusted the scores in our table accordingly and included this discussion of our results in our revised paper.
>
> **W3 Generalizability and Validation**: Generalizing from synthetic to real-world data is an important direction, thanks for pointing this out. Before discussing our rebuttal experiments, we would like to emphasize that applying models across datasets holds challenges due to non-overlapping emotion categories. We explain more details in the appendix. Based on your suggestion, we have applied our EmoNet-Large model on two real-world datasets, demonstrating strong generalization. In more detail, as shown in the table below, our EmoNet-Face models achieve competitive performance on established datasets: **78.29% mean accuracy on FERD and 75.72% on AffectNet across basic emotion categories when mapping our full 40-emotion categories to their 8 base categories** (see table 1 below). These results show that models trained on EmoNet-Face datasets generalize effectively to real-world scenarios. For some categories, the alignment is even near 100% like happiness or surprise. The strong performance despite mapping limitations suggests robust synthetic-to-real transfer. Additionally, synthetic data offers significant advantages: it enables ethical study of sensitive emotions at scale, eliminates confounders present in real-world imagery, and provides increasing ecological validity as synthetic content becomes prevalent online. We have added these results with a new paragraph on synthetic-to-real generalization to the paper and extended our discussion.
>
> **W4 Static vs. Dynamic Datasets**: While dynamic information is valuable, fine-grained recognition from static faces remains an unsolved problem, as our benchmark demonstrates. Our models can already be applied to videos through per-frame predictions, and EmoNet-Face provides a controlled foundation for future dynamic extensions. We view this as an exciting direction for future work and have extended this part of our paper accordingly.
>
>
> We believe these clarifications have addressed your concerns. We are happy about further discussion on these matters.
>
>
> | **Emotion**    | **FERD Acc. (%)** | **AffectNet Acc. (%)** |
> |------------|----------------|---------------------|
> | Anger      | 73.68          | 77.05               |
> | Contempt   | 31.58          | 28.75               |
> | Disgust    | 78.95          | 40.53               |
> | Fear       | 100.00         | 69.08               |
> | Happy      | 100.00         | 99.25               |
> | Neutral    | 84.21          | 78.96               |
> | Sad        | 78.95          | 83.94               |
> | Surprise   | 78.95          | 98.70               |
> | **Mean Acc.** | **78.29**     | **75.72**           |
> **Table 1**: EmoNet-Face Small Model performance on the real-world emotion datasets Facial Emotion Recognition Dataset (FERD) and AffectNet.

---

> > ### Comment · Reviewer_WZ4Z · 2025-08-05
> >
> > The response addressed most of my concerns, and I agree that manual expert curation was essential to create a gold-standard benchmark.

---

> ### Author Response · Authors · 2025-08-05
>
> We thank the reviewer for the response and engagement. We are happy to have addressed the open concerns and kindly ask to consider revisiting the score, and we remain happy to clarify any remaining concerns.

---

### Official Review · Reviewer_SRgU · 2025-07-02

**Rating:** 4
**Confidence:** 4

**Summary:**

The paper introduces the EmoNet-Face dataset, a benchmark dataset annotated by emotion experts that enables fine-grained facial emotion recognition using synthetically generated images. Based on foundational literature in psychology, the authors establish a classification system encompassing 40 emotion categories, far surpassing existing discrete emotion labels. They release three components: a pretraining dataset (EmoNet-Face Big), a fine-tuning dataset (EmoNet-Face Binary), and an evaluation benchmark (EmoNet-Face HQ).

The dataset was constructed with careful consideration of demographic balance and is expert-annotated, ensuring high reliability. Furthermore, the proposed model EmpathicInsight-FACE, built upon a SigLip2 + MLP architecture, demonstrates performance comparable to that of human annotators when trained on the dataset.

**Dataset Code Accessibility:**

Partly

**Dataset Code Comments:**

The manuscript states that the dataset, including the generation prompts and annotation tool, will be made publicly available. While the dataset construction process is described in detail from a reproducibility standpoint, access to the actual dataset is currently limited in both the main text and the appendix.

**Ethical Considerations:**

No, there are no or only very minor ethics concerns

**Limitations Weaknesses:**

Limited Consideration of Distribution Gap (Real ↔ Synthetic)
This study constructs and evaluates the dataset and models solely using facial images synthesized via T2I models. As such, it tends to overlook the domain gap between synthetic and real-world facial data. Although this issue is briefly mentioned in Section 5, the paper does not include any experiments that assess how well models trained on synthetic data generalize to “in-the-wild” scenarios.

Lack of Evaluation on Conventional FER Datasets
In a similar vein, the paper does not investigate whether models trained on EmoNet-Face generalize well to established FER datasets such as AffectNet or RAF-DB. Without such evaluations, it remains unclear how much practical contribution the proposed dataset can make to the broader FER community.

Unclear Benchmarking Strategy
The proposed model, EmpathicInsight-FACE, is a relatively simple architecture consisting of SigLIP2 with additional MLP layers. The authors state that the model is used to predict continuous emotion scores on a 0–7 scale, but the rationale behind this specific experimental setting is insufficiently explained. It would strengthen the paper to include additional experiments that directly evaluate classification performance on the proposed 40 fine-grained emotion categories.

**Strengths Contributions:**

Expanded Emotion Label Set
The paper aptly identifies the limitations of existing Facial Expression Recognition (FER) datasets, which typically cover only a small set of basic emotions. Drawing on psychological literature such as the Handbook of Emotions and utilizing expert annotators, the authors propose a fine-grained taxonomy of 40 distinct emotion categories. The construction process further enhances reliability by applying OCR and LLM-based keyword clustering techniques to the psychological texts, followed by expert-guided refinement.

Use of Diverse Text-to-Image (T2I) Models
In generating synthetic facial images, the authors employed various state-of-the-art T2I models, ensuring demographic diversity in terms of gender, age, and ethnicity (ref. Fig. 2). This approach directly addresses and mitigates common issues of data bias and imbalance present in prior FER datasets.

Ensuring Annotation Reliability
As shown in Table 2, the annotation process is systematic and involves multiple stages of refinement, which significantly contributes to the dataset’s reliability. High agreement among annotators from diverse racial backgrounds further substantiates the dataset’s trustworthiness. Moreover, the high agreement rates in contrastive batches underscore the importance of reliably securing negative samples. Figure 2 also demonstrates the dataset's thoughtful consideration of fairness from a demographic perspective, adding to its research value.

---

> ### Author Rebuttal · Authors · 2025-07-31
>
> We thank Reviewer SRgU for their thorough and constructive review, and for appreciating the novelty of our 40-category taxonomy, the diverse T2I model usage, and the reliability of our annotation process.
>
> **W1 Regarding the Synthetic-to-Real Gap and Lack of Evaluation on Conventional FER Datasets**: Generalizing from synthetic to real-world data is an important direction, thanks for pointing this out. Before discussing our rebuttal experiments, we would like to emphasize that applying models across datasets holds challenges due to non-overlapping emotion categories. We explain more details in the appendix. Based on your suggestion, we have applied our EmoNet-Large model on two real-world datasets, demonstrating strong generalization. In more detail, as shown in the table below, our EmoNet-Face models achieve competitive performance on established datasets: 78.29% mean accuracy on FERD and 75.72% on AffectNet across basic emotion categories when mapping our full 40-emotion categories to their 8 base categories. These results show that models trained on EmoNet-Face datasets generalize effectively to real-world scenarios. For some categories, the alignment is even near 100% like happiness or surprise. The strong performance despite mapping limitations suggests robust synthetic-to-real transfer. Additionally, synthetic data offers significant advantages: it enables ethical study of sensitive emotions at scale, eliminates confounders present in real-world imagery, and provides increasing ecological validity as synthetic content becomes prevalent online. We have added these results with a new paragraph on synthetic-to-real generalization to the paper and extended our discussion.
>
> **W2 Regarding the Benchmarking Strategy**: We appreciate the request for clarification on our architectural choices. The relative simplicity of EmpathicInsight-Face (SigLIP2 + MLP) is actually a deliberate choice, which we consider a strength. This design demonstrates that strong performance on fine-grained emotion recognition can be achieved with straightforward architectures when paired with robust embeddings (e.g. SigLIP) and high-quality data (EmoNet). The simplicity also enables easy adaptation and integration into existing systems. We chose continuous regression (0-7 scale) over classification for two key reasons: (1) Capturing Intensity: Emotions have gradations - regression naturally distinguishes between mild amusement and hysterical laughter, nuances lost in classification; (2) Multi-Label Representation: Faces often display emotion blends. Our approach allows predicting simultaneous high 'Sadness' and moderate 'Disappointment,' providing richer, more realistic affect representation than single-label classification. We have made those design choices clearer in our paper.
>
> **W3 Regarding Dataset Accessibility**: We appreciate your attention to accessibility and acknowledge that we could have been clearer about the included materials. While we didn't include a zip file in the main submission, we provide comprehensive access through anonymized links as noted in footnote 1. Our HuggingFace collection (see dataset URL in the submission) contains all three dataset subsets, and our GitHub repository (also available in the submisison) includes models, training scripts, and Colab notebooks. The generation prompts are included in the dataset (“prompt” column) on the HuggingFace repository as well as our open-source annotation tool that is located in the “annotation-platform” directory in our github repository. In addition, in Appendix A and Figures 10-16, we have provided images describing our annotation process and the annotation interface. We would appreciate it if the reviewer would re-check those already available resources. Furthermore, we have improved the writing of our resources and their accessibility.
>
> We believe these clarifications strengthen our contributions and address the reviewer's concerns. We are looking forward to further discussion on these matters.
>
> | **Emotion**    | **FERD Acc. (%)** | **AffectNet Acc. (%)** |
> |------------|----------------|---------------------|
> | Anger      | 73.68          | 77.05               |
> | Contempt   | 31.58          | 28.75               |
> | Disgust    | 78.95          | 40.53               |
> | Fear       | 100.00         | 69.08               |
> | Happy      | 100.00         | 99.25               |
> | Neutral    | 84.21          | 78.96               |
> | Sad        | 78.95          | 83.94               |
> | Surprise   | 78.95          | 98.70               |
> | **Mean Acc.** | **78.29**     | **75.72**           |
> **Table 1**: EmoNet-Face Small Model performance on the real-world emotion datasets Facial Emotion Recognition Dataset (FERD) and AffectNet.

---

> > ### Author Response · Authors · 2025-08-05
> >
> > We kindly ask reviewer SRgU to consider and engage with our response before the period ends. Thanks!

---

### Official Review · Reviewer_sLpe · 2025-07-02

**Rating:** 5
**Confidence:** 4

**Summary:**

This paper presents EmoNet-Face, a benchmark suite for fine-grained facial emotion recognition. It includes a 40-category expert-designed taxonomy, three synthetic datasets with controlled demographic diversity, and a baseline model (EMPATHICINSIGHT-FACE) that performs near human level. All resources will be publicly released.

**Dataset Code Accessibility:**

Yes

**Dataset Code Comments:**

All datasets, code, and models will be released on HuggingFace and GitHub with documentation, making the work reproducible and accessible.

**Ethical Considerations:**

No, there are no or only very minor ethics concerns

**Final Justification:**

I have read the rebuttal and discussion carefully. I maintain my score to 5:Accept.

**Limitations Weaknesses:**

- The benchmark only uses synthetic images; generalization to real-world data is unclear.

- Some emotions show low annotation agreement, highlighting the subjectivity of labels.

- The taxonomy may not fully reflect cross-cultural emotion understanding.

**Strengths Contributions:**

- Introduces a much richer emotion taxonomy than existing benchmarks.

- Provides large-scale, expert-annotated synthetic datasets with diverse demographics.

- Annotation process is rigorous, with solid inter-rater agreement analysis.

- EMPATHICINSIGHT-FACE achieves strong performance, outperforming current VLMs.

- The paper is well-written, and the release plan is comprehensive and reproducible.

---

> ### Author Rebuttal · Authors · 2025-07-31
>
> We are grateful to Reviewer sLpe for their review and helpful criticism. We appreciate the positive assessment and recognition of the value of our rich taxonomy, rigorous expert-annotated datasets, strong baseline model, and comprehensive release plan. We have addressed the reviewer’s considerations below:
>
> **W1 Synthetic-to-real Data**: Generalizing from synthetic to real-world data is an important direction, thanks for pointing this out. Before discussing our rebuttal experiments, we would like to emphasize that applying models across datasets holds challenges due to non-overlapping emotion categories. We explain more details in the appendix. Based on your suggestion, we have applied our EmoNet-Large model on two real-world datasets, demonstrating strong generalization. In more detail, as shown in the table below, our EmoNet-Face models achieve competitive performance on established datasets: 78.29% mean accuracy on FERD and 75.72% on AffectNet across basic emotion categories when mapping our full 40-emotion categories to their 8 base categories. These results show that models trained on EmoNet-Face datasets generalize effectively to real-world scenarios. For some categories, the alignment is even near 100% like happiness or surprise. The strong performance despite mapping limitations suggests robust synthetic-to-real transfer. Additionally, synthetic data offers significant advantages: it enables ethical study of sensitive emotions at scale, eliminates confounders present in real-world imagery, and provides increasing ecological validity as synthetic content becomes prevalent online. We have added these results with a new paragraph on synthetic-to-real generalization to the paper and extended our discussion.
>
> **W2 Label Subjectivity**: We thank the reviewer for pointing out the crucial issue of ambiguous facial emotions. We totally agree that there are challenges for evaluating emotions, and some may overlap, as discussed in Section 5. Yet, we argue that the lower agreement for subtle emotions reflects the inherent subjectivity of emotion perception–a central finding of modern affective science (Section 3, lines 174-178). Our expert annotators show strong agreement on clear emotions but naturally diverge on ambiguous ones, demonstrating that EmoNet-Face captures genuine psychological complexity. A benchmark with 100% agreement across all emotions is unrealistic and leads to non-robust AI systems regarding emotions. We have updated our discussion accordingly and emphasized this point more.
>
> **W3 Cross-Cultural Understanding**: We fully agree, acknowledge this important limitation (Section 5, line 311), and view it as an exciting direction for future research. By explicitly controlling for diverse ethnicities in our generation prompts, we provide a foundation that can support cross-cultural investigations as the field advances.
>
> We thank the reviewer for their positive assessment and constructive feedback. We appreciate further discussing the points raised.
>
> | **Emotion**    | **FERD Acc. (%)** | **AffectNet Acc. (%)** |
> |------------|----------------|---------------------|
> | Anger      | 73.68          | 77.05               |
> | Contempt   | 31.58          | 28.75               |
> | Disgust    | 78.95          | 40.53               |
> | Fear       | 100.00         | 69.08               |
> | Happy      | 100.00         | 99.25               |
> | Neutral    | 84.21          | 78.96               |
> | Sad        | 78.95          | 83.94               |
> | Surprise   | 78.95          | 98.70               |
> | **Mean Acc.** | **78.29**     | **75.72**           |
> **Table 1**: EmoNet-Face Small Model performance on the real-world emotion datasets Facial Emotion Recognition Dataset (FERD) and AffectNet.

---

> > ### Comment · Reviewer_sLpe · 2025-08-05
> >
> > Thanks for your detailed answer, and I have read carefully. I will submit my score.

---

> > > ### Author Response · Authors · 2025-08-05
> > >
> > > Thank you for your engagement and for reviewing our response!

---

### Official Review · Reviewer_fJNv · 2025-07-07

**Rating:** 4
**Confidence:** 3

**Summary:**

The paper introduces EMONET-FACE, a new benchmark suite for fine-grained facial emotion recognition. It addresses the limitations of existing datasets—such as narrow emotional coverage, poor annotation quality, and lack of demographic diversity—by introducing a 40-category emotion taxonomy, developed from psychological literature and expert input. The paper aims to set a new standard for evaluating and training emotion recognition systems, particularly for applications involving synthetic facial imagery.

**Additional Feedback:**

The Author has partially resolved my confusion.

**Dataset Code Accessibility:**

Yes

**Dataset Code Comments:**

The dataset is available, and the address of the dataset is provided in the paper: https://huggingface.co/collections/t1a5anu-anon/emonet-face-6825a1dd6c6ea537cecba7b8

**Ethical Comments:**

There are no ethical issues, but how to resolve the sim-to-real problem is something that needs to be considered.

**Ethical Considerations:**

No, there are no or only very minor ethics concerns

**Final Justification:**

Thank you for the detailed response. The additional real-world evaluations effectively demonstrate the model’s generalization ability, and the explanation regarding low inter-rater agreement is well-reasoned. Overall, the rebuttal is clear and convincing, and it strengthens the contribution of the paper.

**Limitations Weaknesses:**

1. The dataset consists entirely of synthetic images. It remains unclear how models trained on EMONET-FACE would generalize to real-world, unconstrained imagery.

2. The dataset focuses solely on facial expressions without including contextual or multimodal cues, which are critical for accurate emotion interpretation.

3. Some emotion categories are inherently hard to distinguish visually, leading to low inter-rater agreement for certain classes.

**Strengths Contributions:**

1. The 40 emotion categories go far beyond standard models, including subtle states like shame, fatigue, and intoxication. The taxonomy is backed by literature and refined by psychological experts.

2. Unlike crowdsourced datasets, EMONET-FACE uses trained annotators with psychology backgrounds, leading to higher inter-rater reliability and more trustworthy labels.

3. Images were generated with careful prompt engineering using multiple T2I models, with demographic balance across age, gender, and ethnicity.

---

> ### Author Rebuttal · Authors · 2025-07-31
>
> We sincerely thank Reviewer fJNv for their review and helpful feedback. We appreciate the positive assessment and recognition of the key strengths of our work, particularly our novel 40-category taxonomy, expert annotators, and demographic balancing.
>
> **W1 Synthetic-to-Real Gap**: Generalizing from synthetic to real-world data is an important direction, thanks for pointing this out. Before discussing our rebuttal experiments, we would like to emphasize that applying models across datasets holds challenges due to non-overlapping emotion categories. We explain more details in the appendix. Based on your suggestion, we have applied our EmoNet-Large model on two real-world datasets, demonstrating strong generalization. In more detail, as shown in the table below, our EmoNet-Face models achieve competitive performance on established datasets: 78.29% mean accuracy on FERD and 75.72% on AffectNet across basic emotion categories when mapping our full 40-emotion categories to their 8 base categories. These results show that models trained on EmoNet-Face datasets generalize effectively to real-world scenarios. For some categories, the alignment is even near 100% like happiness or surprise. The strong performance despite mapping limitations suggests robust synthetic-to-real transfer. Additionally, synthetic data offers significant advantages: it enables ethical study of sensitive emotions at scale, eliminates confounders present in real-world imagery, and provides increasing ecological validity as synthetic content becomes prevalent online. We have added these results with a new paragraph on synthetic-to-real generalization to the paper and extended our discussion.
>
> **W2 Lack of Contextual and Multimodal Cues**: EmoNet-Face is designed to benchmark facial emotion recognition in isolation, providing a robust foundation for future multimodal systems. By focusing exclusively on faces, we eliminate contextual confounders and measure pure facial emotion recognition capabilities. This approach aligns with the Theory of Constructed Emotion (doi:10.1093/scan/nsw154) while providing a clean, standardized evaluation of this critical non-verbal communication channel. We agree that future work should focus on multimodal emotion understanding, perhaps using our benchmark as a starting point. We have added this discussion to our paper.
>
> **W3 Low Inter-Rater Agreement for Some Categories**: We thank the reviewer for pointing out the crucial issue of ambiguous facial emotions. We totally agree that there are challenges for evaluating emotions, and some may overlap, as discussed in Section 5. Yet, we argue that the lower agreement for subtle emotions reflects the inherent subjectivity of emotion perception–a central finding of modern affective science (Section 3, lines 174-178). Our expert annotators show strong agreement on clear emotions but naturally diverge on ambiguous ones, demonstrating that EmoNet-Face captures genuine psychological complexity. A benchmark with 100% agreement across all emotions is, in our opinion, unrealistic, lacks demographic diversity, and leads to non-robust AI systems regarding emotions. We have updated our discussion accordingly and emphasized this point more prominently.
>
> We thank the reviewer for their valuable feedback and appreciate further discussion on these matters.
>
> | **Emotion**    | **FERD Acc. (%)** | **AffectNet Acc. (%)** |
> |------------|----------------|---------------------|
> | Anger      | 73.68          | 77.05               |
> | Contempt   | 31.58          | 28.75               |
> | Disgust    | 78.95          | 40.53               |
> | Fear       | 100.00         | 69.08               |
> | Happy      | 100.00         | 99.25               |
> | Neutral    | 84.21          | 78.96               |
> | Sad        | 78.95          | 83.94               |
> | Surprise   | 78.95          | 98.70               |
> | **Mean Acc.** | **78.29**     | **75.72**           |
> **Table 1**: EmoNet-Face Small Model performance on the real-world emotion datasets Facial Emotion Recognition Dataset (FERD) and AffectNet.

---

> > ### Comment · Reviewer_fJNv · 2025-08-03
> > **Reply to author**
> >
> > Thank you for your reply. I have reviewed your answer and will submit my score.

---

> > > ### Author Response · Authors · 2025-08-03
> > >
> > > Thank you for your engagement and for reviewing our response. We're happy to clarify any remaining points.

---

### Comment · Reviewer_fJNv · 2025-08-08

The author's dataset does not raise any ethical issues. Although the proposed dataset is a synthetic dataset, it also performs well in real-world testing.

---

> ### Author Response · Authors · 2025-08-08
>
> We thank the reviewer for confirming no ethical concerns around our paper.

---

### Note · Authors · 2025-08-12

Dear Reviewers and Area Chairs,

We would like to thank you all for your feedback and engagement during the review process, which has significantly helped us improve our work.

We are particularly pleased that our **new rebuttal experiments, demonstrating the effective generalization of our models to real-world datasets**, addressed your primary concerns about the synthetic-to-real gap. We also appreciate the positive feedback on the core contributions of our paper: the novel 40-category emotion taxonomy, the rigorous expert-annotation process of our synthetic, multicultural face datasets, and the strong performance of our EmpathicInsight-Face models. The ethics reviews also strengthened our discussion on responsible use. We believe EmoNet is a valuable and robust benchmark for developing more nuanced and empathetic AI systems and research.

Thank you again for your time and expertise.

---

### Decision · Program_Chairs · 2025-09-18

**Decision:**

Accept (poster)

**Comment:**

I thank the reviewers for their careful and constructive reviews and for engaging in discussion with the authors.

The paper was recognized for addressing limitations of existing emotion recognition benchmarks. Reviewers appreciated the introduction of a fine-grained 40-category taxonomy, the rigorous use of expert annotators, demographic balance in the synthetic data, and the performance of the proposed model.

All the reviewers were initially concerned about the dataset's use of fully synthetic imagery and how that would transfer to real-world conditions. They further noted low annotation agreement for some categories due to the inherent subjectivity of emotions (fJNv, sLpe), and questioned whether the taxonomy fully captured cross-cultural variation (sLpe). Other concerns included limited evaluations and the absence of temporal emotional cues.

In their rebuttal, the authors shared new results showing strong generalization to real-world datasets and clarified their design decisions, licensing, and ethical safeguards. They also addressed concerns about annotation subjectivity, cross-cultural scope, and dataset accessibility. Reviewers generally found these responses detailed and convincing. The authors’ replies seemed to have satisfied the reviewers.

The reviewers' final recommendations were Accept (sLpe) and fJNv, SRgU, and WZ4Z borderline accept.

As for the ethics checks, there were initial concerns about licensing of generated images and possible misuse in surveillance or manipulation. The authors clarified ownership rights, tied usage to regulatory compliance, and added guidelines to restrict use in sensitive domains. These clarifications seemed to address the outstanding concerns.

Having read the paper, the reviews and discussions, I recommend accepting the paper. This set can be a valuable resource, especially in light of the care given to balancing images across demographics and the reduced risk associated with AI generated images.

===== FINAL UPDATE FROM DB Track PCs ====

The final decision for this paper has been taken by the program chairs after consultation with the SACs. All Senior Area Chairs have ranked papers according to the feedback from the AC during the review process. We decided to leave the original meta-review to reflect the opinion of the AC in light of the initial discussions with reviewers and SAC.